# The Electrical Activity of Saharan Dust as perceived from Surface Electric Field Observations

Vasiliki Daskalopoulou[1,2], Sotirios A. Mallios[2], Zbigniew Ulanowski[3,4], George Hloupis[5], Anna Gialitaki[2,6], Ioanna Tsikoudi[2,7], Konstantinos Tassis[8,9] and Vassilis Amiridis[2]

[1]Department of Physics, Faculty of Astrophysics and Space Physics, University of Crete, Heraklion GR-70013, Greece
[2]Institute for Astronomy, Astrophysics, Space Applications and Remote Sensing, National Observatory of Athens, Athens GR-15236, Greece
[3]Department of Earth and Environmental Sciences, University of Manchester, Manchester M13 9PL, UK
[4]British Antarctic Survey, NERC, Cambridge CB3 0ET, UK
[5]Department of Surveying and GeoInformatics Engineering, University of West Attica, Aegaleo Campus GR-12244, Greece
[6]Laboratory of Atmospheric Physics, Department of Physics, Aristotle University of Thessaloniki, Thessaloniki GR-54124, Greece
[7]Department of Environmental Physics and Meteorology, National and Kapodistrian University of Athens, Athens, Greece
[8]Department of Physics, and Institute for Theoretical and Computational Physics, University of Crete, Heraklion GR-70013, Greece
[9]Institute of Astrophysics, Foundation for Research and Technology-Hellas, Heraklion GR-71110, Greece

*Correspondence to*: Vassilis Amiridis (vamoir@noa.gr)

**Abstract.** We report on the electric field variations during Saharan dust advection over two atmospheric remote stations in Greece, using synergistic observations of the vertical atmospheric electric field strength ($E_z$) at ground level and the lidar-derived particle backscatter coefficient profiles. Both parameters were monitored for the first time with the simultaneous deployment of a ground-based field mill electrometer and a multi-wavelength polarization lidar. The field mill timeseries are processed to extract the diurnal variations of the Global Electric Circuit and remove fast field perturbations due to peak lightning activity. In order to identify the influence of the elevated dust layers on the ground $E_z$, we extract a Localized Reference Electric Field from the timeseries that reflects the local fair weather activity. Then, we compare it with the reconstructed daily average behaviour of the electric field and the Saharan dust layers' evolution, as depicted by the lidar. The observed enhancement of the vertical electric field (up to ~ 100 V/m), for detached pure dust layers, suggests the presence of in-layer electric charges. Although higher dust loads are expected to result in such an electric field enhancement, episodic cases that reduce the electric field are also observed (up to ~ 60 V/m). To quantitatively approach our results, we examine the dependency of $E_z$ against theoretical assumptions for the distribution of separated charges within the electrified dust layer. Electrically neutral dust is approximated by atmospheric conductivity reduction, while charge separation areas within electrically active dust layers are approximated as finite extent cylinders. This physical approximation constitutes a more realistic description of the distribution of charges, as opposed to infinite extent geometries, and allows for analytical solutions of the electric field strength, so that observed variations during the monitored dust outbreaks can be explained.

**Keywords:** Dust Electrification; Atmospheric Electric Field Measurements; Reference Electric Field; Charge Separation;

## 1 Introduction

The Global Electric Circuit (GEC) represents the electric current pathway in the Earth's atmosphere. The electric current that flows upwards from thunderstorms and electrified clouds into the Ionosphere, spreads out over the globe along magnetic field
lines to the opposite hemisphere, and returns to the surface of the Earth as the fair weather air-to-Earth current (Bering et al., 1998). The GEC is established by the conducting atmosphere sandwiched between the conductive Earth and the conductive Mesosphere/Ionosphere (Williams, 2009). Atmospheric electric parameters, such as the vertical Electric Field ($E_z$) and induced air-to-Earth current ($I_c$) through the GEC, greatly depend on ambient weather conditions and convective meteorological systems (Kourtidis et al., 2020) due to the re-distribution of charged or uncharged aerosols and terrestrial radioactive particles
in the Earth's atmosphere (Harrison and Ingram, 2005; Wright, 1933). Under fair weather conditions, which are defined according to international standards as those with cloudiness less than 0.2, wind speed less than 5 m/s and the absence of fog or precipitation (Chalmers, 1967; Harrison and Nicoll, 2018), the atmospheric electrical circulation is dominated by the potential difference between the global capacitor planes (about 250 kV, e.g., Rycroft et al., 2008), which in turn generates the fair weather electric field, and consequently the fair weather electric current in the presence of the conducting atmosphere. An
average current density of 2 pA/m$^2$ and a downward looking (by convention positive, e.g., Rakov and Uman, 2003, pp.8) electric field equal to a typical value of about 130 V/m are expected, respectively (Rycroft et al., 2008). The daily variation of the global thunderstorm activity modulates the electric field strength and the resulting diurnal variation is represented by the Carnegie curve (Harrison, 2013).

Amongst the aerosols affecting the atmospheric electrical content (Whitby and Liu, 1966), mineral dust represents one of the
most significant contributors, along with volcanic ash (Harrison et al., 2010), due to its mineralogical composition that results in different electrical properties of the dust particles (Kamra, 1972) and its abundance in terms of dry mass (Tegen et al., 1997). During dust storms, dust devils and subsequent advection of elevated dust layers the electrical parameters can vary greatly from the values under fair weather conditions (Harrison et al., 2016; Renno and Kok, 2008; Zheng, 2013). It is well documented that over deserts the emission process of dust particles can generate large atmospheric electric fields (Esposito et al., 2016;
Renno and Kok, 2008; Zheng, 2013) that affect their flow dynamics (Kok and Renno, 2006). Charged dust occurrences are recorded via ground-based methods also in destinations further away from the source (Harrison et al., 2018; Katz et al., 2018; Silva et al., 2016; Yair et al., 2016; Yaniv et al., 2017), while balloon-borne observations (Kamra, 1972; Nicoll et al., 2011) indicate that space charge is indeed persistent within lofted dust layers during their transport to long distances. The exact mechanisms that would explain and sufficiently describe the long-range electrification of dust are not clear yet, and remain
under investigation. Major processes that are considered responsible for the electrification of dust particles include ion attachment (Tinsley and Zhou, 2006) and particle-to-surface or particle-to-particle collisions, i.e. triboelectrification (Kamra, 1972; Lacks and Shinbrot, 2019; Waitukaitis et al., 2014). Such processes are claimed to have large impact on desert dust transport and its influence in climate and ecosystems through the retention of larger dust particles in the atmosphere (van der

Does et al., 2018; Ryder et al., 2018), as well as to particle vertical orientation with impact on radiative transfer (Bailey et al., 2008; Mallios et al., 2021; Ulanowski et al., 2007).

Ground-based electric field measurements can be indicative of the electrical behaviour of elevated dust layers. These measurements can provide useful information if they are combined with other retrievals on aerosol profiling (e.g. lidar, ceilometer) (Nicoll et al., 2020). However, features of E-field timeseries, such as the enhancement of the near-ground electric field during dust outbreaks, are still unexplained in broad literature (Yaniv et al., 2016, 2017). Observations of enhanced or even reversed E-field at the height of the ground-based sensor, e.g. an electrostatic fieldmeter, are attributed by Ette (1971) and Freier (1960) to charge separation within electrically active dust. According to several laboratory studies (Duff and Lacks, 2008; Forward et al., 2009; Inculet et al., 2006; Waitukaitis et al., 2014), charge transfer processes lead to smaller particles being negatively charged while larger particles tend to be positively charged, therefore charge separation within lofted dust layers is also possible due to the expected size selective gravitational settling that could stratify the fine and coarse mode particles (Ulanowski et al., 2007). An observed reduction of the E-field in a mountainous area is attributed to the superposition of two dust layers in different heights with respect to the ground-based sensor (Katz et al., 2018). Moreover, layers that exhibit large particle densities lead to more particles competing for the same amount of ions (ion-particle competition, e.g., Gunn, 1954; Reiter, 1992), hence they act as a passive element within the atmospheric circulation and can reduce the near-ground electric field. A similar reduction of the electric field can be expected whenever, for any reason, the charge separation does not occur. As an example, one can think meteorological conditions that force the particles to move randomly, cancelling their vertical movement and, therefore, the charge separation. Nonetheless, systematic profiling measurements are needed so as to fully characterize the electrical properties of the dust particles aloft, with respect to the locally occurring meteorological conditions.

In this study, we focus on monitoring perturbations of the E-field near the ground caused by the transported dust layers, with special emphasis on slow E-field perturbations (with duration larger than 6 hours to exclude phenomena with small timescales or local effects of random origin), and we attempt to classify and comment on the electrical activity of the dust layers. As electrically active we define the layers that exhibit charge separation and behave as electrostatic generators in the GEC, similarly to electrified shower clouds and thunderstorms (e.g., Mallios and Pasko, 2012). Conversely, electrically neutral are assumed to be the layers with no charge separation which, therefore, act as passive elements in the GEC, similarly to the non-electrified shower clouds (e.g., Baumgaertner et al., 2014). Four selected cases of Saharan dust plumes are examined, as captured over Finokalia and Antikythera atmospheric observatories by the same ground-based electrometer, as well as by the sophisticated Polly[XT] lidar system. In Section 2, we provide an overview of the instrumentation and measurement techniques, and specify the methods used to parameterize the electrical behaviour of the dust layers. In Section 0, we present the modelled E-field behaviour which is used as a proof of concept for the explanation of the E-field diurnal variation (relative to the local reference field), presented in the results section along with the dynamic evolution of the dust episodes as revealed by the profiling information from the lidar. We further discuss whether the configuration of finite cylindrical charge accumulation

regions, previously suggested for the representation of charge distributions within thunderclouds (Krehbiel et al., 2008; Riousset et al., 2007), is capable of reproducing our experimental results. Finally, we present our conclusions in Section 6.

## 2 Data and methodology

We analyse four Saharan dust outbreaks recorded over two observational sites in Greece. The first atmospheric monitoring station is situated in the remote location of Finokalia (35.338° N, 25.670° E) on the north eastern coast of Crete, with the nearest large urban center being the city of Heraklion located 70 km to the west. The station is located at the top of a hill (252 m asl) facing the sea within a sector of 270° to 90° and the climatic characteristics are typical of the eastern Mediterranean basin exhibiting two distinctive seasons, the dry season (April to September) characterized by increased levels of pollution

and biomass burning and the wet season (October to April). Significant Saharan dust transport occurs when S/SW winds are prevalent during the intermediate season of March till June and may lead to ground concentrations exceeding 1 mg/m$^3$ (Solomos et al., 2018). Since there is no significant human activity occurring at a distance shorter than 15 km within the above sector, it makes it an appropriate location for monitoring dust layers advected directly from the Sahara. The second site is the PANhellenic GEophysical observatory (PANGEA) in the remote island of Antikythera (35.861° N, 23.310° E, 193 m asl). The

island covers an area of just 20.43 km$^2$, 38 km south-east of the larger island of Kythera and is devoid of human activity as its inhabitants are at most twenty people during early fall to mid-summer. The station location is ideal as the island is placed at a crossroad of air masses (Lelieveld et al., 2002), with NNE winds being prominent between August and February, while in spring and early summer western airflows that favor dust transport are observed. Moreover, the prevailing meteorological conditions on the island are again representative of the eastern Mediterranean with warm and dry days in summer in contrast

to winter, when the days are colder and wetter days are typical. The dust outbreaks recorded were on the 25[th] of July 2017 and March 16[th] 2018 on Finokalia, October 20[th] 2018 and June 23[rd] 2019 on Antikythera, selected due to the presence of elevated dust layers in the lidar profiles.

### 2.1 Aerosol monitoring and characterization

### 2.1.1 Lidar measurements

For the comprehensive characterization of dust particle optical properties, we exploit the profiling capabilities of the Polly[XT] Raman polarization lidar (Engelmann et al., 2016) of the National Observatory of Athens (NOA), as part of the European Aerosol Research Lidar Network (EARLINET). This multi-wavelength system is equipped with three elastic channels at 355, 532 and 1064 nm, two vibrational Raman channels at 387 and 607 nm, two channels for the detection of the cross-polarized backscattered signal at 355 and 532 nm, and one water vapour channel at 407 nm. The system employs two detectors, a near-

field and a far-field telescope provide reliable aerosol optical property profiles from close to the ground to the upper troposphere. The basic lidar quantities used for the monitoring and characterization of dust loads in our study, are the total attenuated backscatter coefficient (Mm$^{-1}$sr$^{-1}$) at 532 nm (calibrated range-corrected signal) to account for particle

concentrations and the Volume Linear Depolarization Ratio (VLDR, $\delta_v$) at 532 nm. VLDR (%) is the ratio of the cross–polarized to the co-polarized backscattered signal (Freudenthaler et al., 2009), where cross- and co- are defined with respect to the plane of polarization of the emitted laser pulses. It encloses the influence of both atmospheric particles and molecules, with high $\delta_v$ values being indicative of irregular particles (i.e. atmospheric dust). However, for a comprehensive aerosol characterization, the particle backscatter coefficient ($\beta$) and Particle Linear Depolarization Ratio (PLDR, $\delta_p$) are needed. PLDR (%) is derived from VLDR by correcting for molecular depolarization with atmospheric parameters extracted from radiosonde measurements (i.e. atmospheric pressure and temperature). In the selected case studies, we also present the $\delta_p$ and $\beta$ profiles, as derived in the timeframe when each dust episode was fully developed (averaged between 18:00 and 21:00 (UTC) for all dust cases). Typical $\delta_p$ values for Saharan dust are in the range of 25% to 35% at 532 nm, while large $\beta$ values are representative of substantial particle concentrations (Haarig et al., 2017; Veselovskii et al., 2016, 2020).

### 2.1.2 Ancillary aerosol and trajectory information

The Aerosol Optical Depth (AOD) was monitored by a CIMEL sunphotometer, part of the Aerosol Robotic Network (AERONET - https://aeronet.gsfc.nasa.gov/), which was co-located with the lidar on both stations. For the cases examined here, the AOD varied from 0.221 to 0.366 at 500 nm. To characterize the air masses in regard to their origin we use the NOAA HYSPLIT back trajectory model, driven by GDAS meteorological data (https://www.ready.noaa.gov/HYSPLIT.php). The arrival heights for dust over the observational sites were selected in HYSPLIT according to the prevailing layering depicted by our lidar measurements (Fig. 1).

### 2.2 Electric Field measurements and data processing

### 2.2.1 Ground-based E-Field measurements

The JCI 131 Field mill (FM) electrometer (Chubb, 2014; Chubb, 2015) was installed in Finokalia from April 2017 until May 2018 (382 days) and then re-located to Antikythera, where the examined timeseries span from June 2018 to June 2019 (243 days) for continuous monitoring of the near-ground (on instrument mast height) vertical electric field. Field mills are robust instruments, mostly used for lightning warning applications providing, though, sufficient sensitivity for the detection of weaker electric fields. The instrument was mounted on a 3 m pole, and as far as possible from physical obstacles, buildings and any metallic objects that could create distortions to the electric field. However, on Finokalia the FM was on the edge of a hilly elevation which added a topography factor, not quantified in the specific research due to the lack of typical flat ground measurements in the area. On Antikythera, the mill installation location could be more carefully selected to avoid orography, obstacles and power grid lines. Instrument output range was set to the most sensitive scale (2.0 kV full scale) with a sensitivity of the order of 1 V/m for 1 Hz measurement frequency and the data were acquired from a 24-bit local data-logger. In order to interpret the field mill measurements, it is essential to compare the data with a reference field representative of local fair weather conditions. The methodology followed for this process is described in the paragraph below.

### 2.2.2 Derivation of the Localized Reference Electric Field

The classification of the vertical electric field behaviour under dust influenced conditions, as that of an enhanced, reduced or reversed E-field, necessitates comparison with the local long-term fair weather electric field. In order to represent solely the diurnal GEC influence at each observational site, away from electric generators perturbing the near ground E-field (e.g., Zhou and Tinsley, 2007), we construct a Localized Reference Electric Field (LREF) by exploiting only the timeseries inherent attributes and the measuring quantity itself, through the processing chain described below (Fig. 2). Various authors have

presented different methodologies for determining fair weather conditions (e.g., Anisimov et al., 2014). For the specific study, the selected constraints of fair weather are based on the classification of fair weather days as the less electrically disturbed days, also assumed by the Carnegie Institute researchers (Harrison, 2013b). Although, local effects on the E-field at each site can be of random nature (wind gusts, lightning strikes, radon emission and turbulent flows due to orography), the selection of fair weather data can be based on noise reduction by subtracting values which are clearly dominated by local influences and

not directly addressing the meteorological criteria of fair weather (Harrison and Nicoll, 2018).

As such, the FM data are pre-processed by applying the appropriate scaling factor for the 3 m mounting mast of the electrometer (Chubb, 2015) and then days with no missing values due to either instrument malfunction, power outages or pc communication failures, are selected (Filter no. 1). Under local fair weather conditions, the E-field, as measured here, is positive therefore imposing the second filtering step with a non-negativity constraint (Filter no. 2). When representing the E-field diurnal

variation by the Carnegie Curve, which is used consistently as a reference against locally measured atmospheric electricity parameters, the hourly variations of the field that shape the curve correspond to the 24, 12, 8, and 6-hour durations, as deduced from previous consistent observations of the Carnegie vessel (Harrison, 2013b). The present study attempts to derive the local harmonic fit in the form of the LREF, based on the Carnegie curve morphology, and assuming that this trend should be followed by the reference field as well. Consequently, the averaged 1s data to 1-minute data (datalogger configuration) are shifted to

the frequency domain through a Fast Fourier Transform (FFT) representation so as to evaluate the relative contributions of the first five principal harmonics to the diurnal cycle of the electric field (hourly variations including daily mean), which are depicted in the following signal equation for $S(t)$ (1). We note, that days with missing data are removed, because the uneven temporal distribution of the measurements modifies the time window for the FFT algorithm, and therefore, modifies the timeseries spectrum.

$$S(t) = A_0 + A_1 \cos(2\pi f_1 + \varphi_1) + A_2 \cos(2\pi f_2 + \varphi_2) + A_3 \cos(2\pi f_3 + \varphi_3) + A_4 \cos(2\pi f_4 + \varphi_4) \tag{1}$$

where $S$ is the electric field at time t in hrs (UTC), $A_i$ for $i = 0,..4$ with $A_0$ representing the mean value (constant, zeroth harmonic) and $A_1$ to $A_4$ (first to fourth harmonic) represent the amplitudes of the 24, 12, 8, and 6-hour variations, $f_i = i \frac{t}{24} 360°$ is the frequency of each harmonic, where $f_0 = 0$ and $\varphi_i$ are the respective phases in degrees, with $\varphi_0 = 0$ (Harrison, 2013). Based on the form of the Carnegie curve, and assuming that this trend should be followed by LREF, we find empirically that the ratio between the zeroth harmonic and the first harmonic is around two. Therefore, the $E_z$ values for which the

amplitude $A_0$ is larger than two times the amplitude $A_1$ are kept (Filter no. 3). The same filter is applied to the other harmonics as well ($A_0$ is larger than two times the $A_i$), making sure that no fast-transient contribution is kept.

Lastly, since the amplitude of each harmonic is expected to be constant for all days (as the amplitudes in the Carnegie curve do), we impose the Chauvenet criterion on each of the filtered five harmonics amplitude, so as to detect outliers. The criterion is imposed once with the use of the relation below:

$$N \, erfc\left(\frac{d^j}{\sqrt{2}s}\right) < \frac{1}{2}$$

(2)

where

$d^j = \left|A_i{}^j - \bar{A}_i\right|$, is the deviation for $i = 0, ..4$ referring to the i$^{th}$ harmonic, $j = 0, ..N$ the day number and N the total number of days, for:

$\bar{A}_i = \frac{1}{N}\sum_{j=1}^{N} A_i{}^j$, where $A_{ij}$ is the i$^{th}$ harmonic amplitude per day and summated over $j$ gives $\bar{A}_i$ as the mean amplitude of each harmonic.

Lastly, $s$ is the unbiased sample variance and is defined as:

$$s^2 = \frac{\sum_{j=1}^{N}\left(A_i{}^j - \bar{A}_i\right)^2}{N - 1}$$

within erfc(x) which is the complementary error function, defined as:

$$erfc(x) = 1 - erf(x) = 1 - \frac{2}{\sqrt{\pi}}\int_0^x e^{-t^2}dt$$

After the Chauvenet criterion is met, 152 total undisturbed weather days are detected for Finokalia and 109 days for Antikythera. From this reduced dataset, we reconstruct the LREF by keeping the mean values of the first three harmonics and calculate the respective standard errors as $\pm 2SE$ from the reconstructed signal.

### 2.2.3 E-field measurement comparison

In order to compare LREF with the daily variation of the electric field during the dust events, these field mill measurements are also shifted to the frequency domain through an FFT. Again, the first five harmonics are retained and from the specific dataset, a smoothed slow varying field is reconstructed (otherwise referred to as reconstructed mean for the remainder of this paper) from the set of mean amplitudes and phases of the first three harmonics. This filtered field retains the main characteristics of the local reference field, since fast transient events which are less than 6 hrs in duration are removed. Therefore, the LREF and reconstructed mean field signals that are compared have the same spectral information. Moreover, to compare the E-field timeseries with the lidar retrievals, all the field mill data are further averaged to 5 mins.

### 2.3 Mathematical formalism for the modelling of the ground E-field

Ideally, under strict fair weather conditions, complete lack of aerosol particles in the atmospheric circulation is expected, since it guarantees that the only mechanism of atmospheric ions loss is the ion-ion recombination. As the concentration of aerosols

increases, additional loss can be due to ions attaching to the particles, which leads to a perturbation of the ion density from fair weather values. In actual conditions, aerosols always exist, but under fair weather conditions their concentrations are small enough to not significantly affect the ionic content of the atmosphere. Therefore, for the modelling purposes of fair weather conditions, aerosol concentrations can be neglected. In the steady state of such an atmosphere, the divergence of the total current is zero $\vec{\nabla}\vec{J}_{tot} = 0$, as a direct consequence of the continuity equation and hence the conduction current remains constant with altitude. From Ohm's law, we can relate the conduction current, $J_z$, with the vertical component of the electric field, $E_z$ (Fig. 3a), as:

$$J_z = \sigma E_z \tag{3}$$

where $\sigma$ is the atmospheric conductivity and assume a smooth conductivity profile along the altitude z, as:

$$\sigma = \sigma_0 \, exp\left(\frac{z}{l}\right) \tag{4}$$

for $\sigma_0$ and $l$ the constants that represent the near ground atmospheric conductivity and the atmospheric scale height, respectively. The given mathematical formalism of the atmospheric conductivity is adopted also by Ilin et al. (2020). The authors demonstrated that such a profile adequately describes the main aspects of the real conductivity distribution, and can be seen as a global mean conductivity profile.

We, then, express the conduction current at ground level, $J_{z_0}$, as a function of the columnar resistance $R_c$ and the potential difference $\Delta V = V_{ion} - V_0$ (5), therefore:

$$J_{z_0} = \frac{\Delta V}{R_c} = \frac{V_{ion}}{R_c} \tag{5}$$

where $V_{ion}$ is the ionospheric potential at the altitude $H$, and $V_0$ is the potential at the Earth's surface which is considered a good conductor due to soil particles that are usually covered by a thin, conducting film of water (Kanagy and Mann, 1994), hence $V_0$ is set equal to zero.

The columnar resistance can be calculated from the conductivity profile of equation (4) (Rycroft et al., 2008), hence:

$$R_c = \int_0^H \frac{dz}{\sigma} = \frac{l}{\sigma_0}\left(1 - exp\left(-\frac{H}{l}\right)\right) \tag{6}$$

By combining equations (3), (5) and (6), the fair weather electric field at ground level, $E_{z_0}$, is of the form (Gringel et al. 1986):

$$E_{z_0} = \frac{V_{ion}}{\sigma_0 R_c} = \frac{V_{ion}}{l\left(1 - exp\left(-\frac{H}{l}\right)\right)} \tag{7}$$

which depends solely on the scale height $l$ and the ionospheric potential $V_{ion}$.

However, the presence of aerosols in the atmosphere and consequently dust particles, affects atmospheric conductivity (Harrison, 2003; Siingh et al., 2007; Tinsley and Zhou, 2006; Zhou and Tinsley, 2007). Aerosols tend to scavenge atmospheric ions due to electrostatic interactions and ion thermal diffusion, leading to a reduction of the atmospheric ion density, and consequently of the atmospheric electrical conductivity. The process of ion attachment to aerosols has been exhaustively investigated in the past literature. A review paper by Long and Yao (2010) contains a summary of all models and theories

regarding the aerosol charging by ions. The case of a steady state atmospheric desert dust layer that does not exhibit charge stratification is examined below. The layer acts as a passive electrical element (resistor), and reduces the fair weather atmospheric conductivity due to the ion attachment to dust particles, by a varying reduction factor $n$. Fig. 3b, represents the above layer configuration, where the new conductivity profile within the layer along the altitude $z$, will be:

$$\sigma' = \frac{\sigma_0}{n} exp\left(\frac{z}{l}\right) \tag{8}$$

The electric field at ground due to the dust layer, $E_{z_0,layer}$, is given by:

$$E_{z_0,layer} = \frac{V_{ion}}{\sigma_0 R_c'} \tag{9}$$

with the new columnar resistance, $R_c'$, being:

$$R_c' = \int_0^{z_1} \frac{dz}{\sigma} + \int_{z_1}^{z_1} \frac{dz}{\sigma'} + \int_{z_1}^{H} \frac{dz}{\sigma}, \qquad z \neq z_{1,2} \quad \Rightarrow$$

$$R_c' = \frac{l}{\sigma_0}\left[1 + (n-1)exp\left(-\frac{z_c - d/2}{l}\right)\left(1 - exp\left(-\frac{d}{l}\right)\right) - exp\left(-\frac{H}{l}\right)\right] \tag{10}$$

where $z_{1,2}$ are the layer bottom/top heights, $z_c$ is the mean layer central height and $d = z_2 - z_1$ the mean layer depth. The dust layer horizontal extent $R$ (radius), as depicted in Fig. 3b, is assumed to be at least ten times larger than its vertical extent ($R \geq 10d$) for a thin layer approximation.

And (9) gives through (10):

$$E_{z_0,layer} = \frac{V_{ion}}{l\left[1 + (n-1)exp\left(-\frac{z_c - d/2}{l}\right)\left(1 - exp\left(-\frac{d}{l}\right)\right) - exp\left(-\frac{H}{l}\right)\right]} \tag{11}$$

Therefore, it is clear that $E_{z_0,layer}$ depends on the scale height parameter $l$, the reduction parameter $n$, the layer central height $z_c$ and the layer depth $d$. A further investigation of the E-field dependence on the various parameters listed above can be found in the Appendix A.

On a next step, we parameterize an electrically active dust layer to calculate its impact on the surface E-field. Specifically, we construct a simplistic model for the atmospheric column (1D), based on the hypothesis that the charge accumulation areas within the dust layer can be approximated by charged cylinders of a total charge density $\pm\rho$ (Fig. 3c). For the cylinder, we assume that its horizontal extent, as represented by the cylinder radius $R_1$ in Fig. 3c, is at least ten times larger than the vertical extent (large cloud approximation), to ensure that the field lines are vertical with only weak radial dependence directly below the center of the layer (e.g., Riousset et al., 2007). The electric field of such an idealized finite extent charged layer is dependent on the distance from the layer. Departures from this behaviour occur near layer edges and distances comparable to the layer extent. Moreover, the hypothesis of the presence of image charges is also applied due to the ground being a good conductor, ensuring that the calculated electric potentials are solutions to the Poisson equation.

The formulation for such an electrically active layer consists of a superposition of the electrically neutral dust layer case with the monopole charged cylinder case, constrained for zero ground and zero ionospheric potentials. The derivation of the ground

electric field due to the presence of a total charge density of $\rho$ is given below. We calculate the potential at point A (central lower point of the charged cylinder), as specified in Fig. 3c, which is given as the sum of the potential from the total charge $Q$ and the potential from its image charge $Q_{img}$, where $Q_{img} = -Q$:

$$V_A = V_Q + V_{Q_{img}} \tag{12}$$

The solution for the potential at the central axis of a solid charged cylinder with total charge density $\rho_1$, is given by (e.g., Griffiths Instructor's Solution Manual for Introduction to Electrodynamics, 4th Edition, 2013):

$$V_Q = \frac{\rho_1}{4\varepsilon_0} \left\{ d_1 \sqrt{R_1^2 + d_1^2} + R_1^2 \ln\left[ \frac{d_1 + \sqrt{R_1^2 + d_1^2}}{R_1} \right] - d_1^2 \right\}, \qquad for\ R_1 \geq 10 d_1 \tag{13}$$

where $\varepsilon_0$ is the permittivity of vacuum, $R_1$ the charge region horizontal extent presented by the cylinder radius, $d_1 = z_2' - z_1'$ the cylinder depth (charge region vertical extent) and $\rho_1$ is the total charge density. Correspondingly, the potential at point A due to the image charge is calculated as:

$$V_{Q_{img}} = \frac{-\rho_1}{4\varepsilon_0} \left\{ 2z_{c_1}\sqrt{R_1^2 + (2z_{c_1})^2} - (2z_{c_1} - d_1)\sqrt{R_1^2 + (2z_{c_1} - d_1)^2} \right.$$

$$\left. + R_1^2 \ln\left[ \frac{2z_{c_1} + \sqrt{R_1^2 + (2z_{c_1})^2}}{(2z_{c_1} - d_1) + \sqrt{R_1^2 + (2z_{c_1} - d_1)^2}} \right] - 2d_1\left(2z_{c_1} - \frac{d_1}{2}\right) \right\} \tag{14}$$

for $z_{c_1}$ the charged area central height. The new columnar resistance up to the height of point A, will be:

$$R_{c_1} = \int_0^{z_1} \frac{dz}{\sigma} + \int_{z_1}^{z_1'} \frac{dz}{\sigma'} \Rightarrow$$

$$R_{c_1} = \frac{l}{\sigma_0}\left[1 - exp\left(-\frac{z_c - d/2}{l}\right)\right] + \frac{nl}{\sigma_0}\left[exp\left(-\frac{z_c - d/2}{l}\right) - exp\left(-\frac{z_{c_1} - d_1/2}{l}\right)\right] \tag{15}$$

calculated from the ground to the dust layer bottom height $z_1$, and from there to the cylindrical charged area bottom height $z_1'$ (Fig. 3c). Note that $d$ is the layer depth while $d_1$ is the cylinder depth.

And again, from Ohm's law and equation (15), we get the electric field at ground level for the case of a charged cylindrical

monopole as:

$$E_{z_0,Q} = \frac{V_A}{l\left[1 - exp\left(-\frac{z_c - d/2}{l}\right)\right] + nl\left[exp\left(-\frac{z_c - d/2}{l}\right) - exp\left(-\frac{z_{c_1} - d_1/2}{l}\right)\right]} \tag{16}$$

where $V_A$ is given from equations (12) to (14), with $E_{z_0,Q}$ being dependent on the scale height $l$, the conductivity reduction factor $n$, the central layer height $z_c$ and the charged area central height $z_{c_1}$.

In the case of multiple stratified charged areas within the layer, the electric field at ground level is a superposition of the contribution to the field from each charge and its image ($E_{z_0,Q_i}$), along with the non-stratified dust layer's contribution attributed to the imposed conductivity reduction ($E_{z_0,layer}$), hence:

$$E_{z_0,multipole} = \sum E_{z_0,Q_i} + E_{z_0,layer} \Rightarrow$$

$$E_{z_0,dipole} = E_{z_0,\text{lower cylinder}} + E_{z_0,\text{upper cylinder}} + E_{z_0,layer}$$

(17)

where, subsequently, if we assume a dipole charge configuration within the dust layer, the total contribution to the ground E-field ($E_{z_0,dipole}$) will be a superposition of the influence from the lower ($E_{z_0,\text{lower cylinder}}$) and upper ($E_{z_0,\text{upper cylinder}}$) charged areas, along with the electrically neutral dust layer's contribution ($E_{z_0,layer}$).

## 3 Model outputs

As a result of the mathematical formalism described in Section 2.3, we present the 1D model outputs and restrictions under which the various behaviours of the near-ground E-field strength can be exhibited in comparison to the calculated fair weather E-field. Following this formulation, the dust layer that exhibits charge separation is approximated with a dipole of oppositely charged cylinders. The influence of small charge imbalances, less than 10%, in the bipolar case, which could quantitatively explain the enhancement or reduction in the E-field is also investigated. If multiple charge accumulation regions are suspected within the dust layer (Zhang and Zhou, 2020), the problem can be still represented by the model output through a superposition of several cylindrical monopoles with different charge densities, polarities and separation distances.

### 3.1 E-field below Fair weather field

In this section, we describe the possible cases under which lofted dust layers can reduce the near-ground E-field strength below the reference electric field values, and we investigate whether electrified dust layers can reproduce such a behaviour. $E_{z_0,layer}$ dependency on the various atmospheric parameters points to atmospheric conductivity as the dominant factor that affects the E-field (see Appendix A). Therefore, we expect that if the dust layer is electrically neutral and acts as a passive element by reducing the atmospheric conductivity, it will greatly affect the field by forcing it below the local reference values.

Since there is little data on vertical profiling of the dust layer electrical properties, we use the previous measurements of electric field variation with altitude, which indicated a charge density of $\rho = \pm 25$ pC/m$^3$ within a transported Saharan dust layer away from the emission source (Nicoll et al., 2011). From the specific value, the total charge $Q$ is estimated for the different model cylinder extents. Gringel and Muhleisen (1978) measured a reduction of the electrical conductivity, compared to the fair weather values, by a factor of four within an elevated dust layer and we, therefore, adopt a reduction factor of $n = 4$ in the present study (see also Appendix A). For $E_{z_0}$, $E_{z_0,layer}$ and $E_{z_0,Q}$ estimations, the scale height is fixed to a globally average value of $l = 6$ km (Kalinin et al., 2014; Stolzenburg and Marshall, 2008), the ionospheric potential is fixed at $V_{ion} = 250$ kV and the ionospheric height is at $H = 70$ km. The mean central height of the dust layer and mean layer depth are both set equal

to 3 km ($z_c = d = 3$ km), since this height represents the average value for the four dust cases according to the lidar PLDR profiles (Table 1).

### 3.1.1 Balanced/Imbalanced dipole field below Fair weather field

We consider the case of two oppositely charged cylinders with similar geometries as in Fig. 3c, assuming they are within a
dust layer with a mean height of 3 km and a mean depth of 3 km. The lower cylinder central height $z_{c_1}$ starts at 2.95 km and decreases, the upper cylinder central height $z_{c_2}$ starts also at 2.95 km for zero separation distance (at this limit, it represents electrically neutral dust that lacks internal E-field generation due to the absence of charge separation) and increases within the dust layer boundaries (varying separation distance), while their depth is fixed at 100 m, in order to be of finite vertical extend but quite thin. The separation distance between the two cylinders is defined as the difference between their central heights and
the ground E-field is a superposition of the electric field of the upper and lower cylinders. We assume the bottom cylinder to be positively charged with density $+\rho$ and the upper one to be negatively charged with $-\rho$ (Fig. 4a), in order to simulate gravitational settling conditions for larger and, most probably, positively charged dust particles (Forward et al., 2009; Waitukaitis et al., 2014). From equations (12) to (17), the field is analytically calculated directly on the axis of the charged cylinders and plotted against the cylinder radius $R$ for separation distances up to 800 m. As seen in Fig. 4, the resulting electric
field values on ground level are consistently below the fair weather constant value. When the dipole separation distance increases, the vertical electric field at ground level increases. This happens due to the stronger influence of the lower charged cylindrical layer to the surface resistance. The fact that the upper charged cylinder moves to higher altitudes signifies that the resistance between the specific layer and the ground increases, therefore the conduction current at the ground decreases. The conduction current due to the upper charged layer, then, becomes weaker than the conduction current due to the lower charged
layer, which moves towards the ground. Since the conductivity at the ground is undisturbed by the dust layer (Fig. 3c) and equal to the fair weather value, the ground electric field due to the upper layer decreases as the layer moves up, while the field due to the lower layer increases as the layer approaches the ground, leading to an increasing value of the total electric field with the increasing separation distance. When the separation distance is kept relatively small, the enhancement effect in the E-field is not significant enough to overcome the fair weather values (Fig. 4). For large radii, although the infinite plates
configuration is asymptotically approached ($E_{z_0,dipole} \rightarrow 0$), there is a nearly-constant residual field for the finite cylindrical geometry of the charged regions. Since the charged cylinders are placed in a conducting medium above a perfect conductor, the electric field at the ground will not be zero even if the cylinders have infinite extent. Due to the conductivity distribution, there is an uneven contribution of the electric fields of each cylinder and, therefore, the E-field is expected to converge to this non-zero value (Fig. 4).
If the dipole charge density is not uniformly distributed to both cylinders, resulting in a charge imbalance within the layer, the electric field will be more sensitive to separation distance changes (Fig. 4b). Such imbalance could be the result of (a) dust charging at the source, prior to any charge separation that may occur (Ette, 1971; Kamra, 1972), (b) charging due to atmospheric current, or (c) charge loss through dry deposition in the Planetary Boundary Layer (PBL). In Fig. 5, the ground

electric field dependence on the separation distance and cylinder radius is depicted, for a charge density difference of $\Delta\rho = 2$ pC/m$^3$ (8%) between the two charged cylindrical areas, with the upper one being less charged. This leads to a larger increase of the E-field than in the balanced dipole case (Fig. 4a), as the effect of the upper cylinder not only decreases as it moves to higher altitudes, but it is also reduced due to the reduction of the total charge density which influences proportionally the electric field. Note that even a small imbalance can highly increase the external field. Nevertheless, for relatively small separation distances the resulting field values fall again below the fair weather value.

## 3.2 E-field above Fair weather field

We examine the physical arrangement within the dust layer that can provide an enhancement to the electric field above the fair weather values and subsequently above the LREF.

### 3.2.1 Balanced/Imbalanced dipole field above Fair weather field

For the same charged region geometries as discussed previously, larger separation distances are imposed for the balanced dipole case (Fig. 5a), but we strictly remain within the base dust layer mean dimensions. Fig. 5 shows that as the separation distance between the oppositely charged layers increases, an enhancement of the E-field above the local reference values occurs. This enhancement becomes more prominent as the layers grow further apart within the dust plume and the contribution from the lower layer is significantly larger than the upper layer. The above dependence of the ground E-field on the separation distance is not expected in the case of charged infinite plates, as discussed in Section 3.1.1. Again, for a charge imbalance of 8% between the two cylinders and for larger separation distances, the E-field is significantly enhanced and exceeds the local fair weather values (Fig. 5b). The term large or small separation distance depends on the conductivity distribution and more specifically on the conductivity scale height, as can be seen in equations (11) and (16). This increase becomes more prominent as the separation distance increases and the lower positive cylinder moves closer to the sensor location.

## 4 Experimental Results

The near ground electric field measurements with co-located lidar observations are presented for the four case studies of elevated Saharan dust layers, over the two atmospheric remote sensing stations. The transient dust events recorded by Polly$^{XT}$ were, simultaneously, electrically monitored throughout the day with the field mill. According to the effect over the E-field timeseries, the dust outbreaks examined are separated into two classes, the ones that effectuate an enhancement to the ground electric field and those inducing a reduction with respect to the local reference field. Through these observations, we attempt to provide evidence of electrically active dust only by ground-based methods, supported by the model configuration described in the previous sections.

## 4.1 Layer characterization through PollyXT

The July 2017 and March 2018 dust events in Finokalia are characterized by large concentrations of airborne dust particles from the middle of the day onwards, followed by dust settling towards the ground after 21:00 (UTC), as indicated by the time-height plots of the total attenuated backscatter coefficient (Fig. 6 and Fig. 8). Larger particle concentrations are shown in red tones, with the β and $\delta_p$ (black lines) profiles superimposed to the respective attenuated backscatter coefficient (top panel) and $\delta_v$ (lower panel) quick-looks. For the first case study (Fig. 6), beta values are between 3 to 4 (Mm$^{-1}$sr$^{-1}$) with a maximum value of 5 (Mm$^{-1}$sr$^{-1}$) inside the layer and denote large particle concentrations. High $\delta_v$ values (> 10%) are indicative of dust particles and $\delta_p$ values between ~ 25% - 30% in the afternoon are characteristic of pure dust. Settling of dust particles below 2 km, inside the Marine Boundary Layer (MABL) is revealed from the time-height evolution of the VLDR (see Fig. 6). For the March 2018 case study (Fig. **8**), the elevated layer (small dust concentration was present near the surface) reached Finokalia at early noon. The layer was directly transported from Sahara and reached the station in less than 48 hours, as indicated by the backward trajectories analysis (Fig. 1c). Examination of the β profile in Fig. 8, shows values that reach up to 15 (Mm$^{-1}$sr$^{-1}$) at the top of the layer, indicating higher aerosol concentrations in this case. $\delta_v$ values close to 30% are indicative of high dust particle concentration and $\delta_p$ values persistently of 30% are characteristic of pure dust within the entirety of the layer (1 to 4 km), with dust downward mixing inside the MABL being less prominent.

The October 20$^{th}$ 2018 Antikythera layer (Fig. 7), exhibits lower dust particle concentrations (β lower than 5 Mm$^{-1}$sr$^{-1}$) close to the ground up to 6 km in altitude, mostly mixed with marine aerosols below 2 km (Fig. 1b and Fig. 7). High $\delta_v$ values (> 20%) are indicative of dust particle presence and $\delta_p$ between 25% - 30% in the afternoon is characteristic of pure dust. It is also observed that the near-ground dust concentration is very low, with the thin layer at 500 m being a mixture of dust particles and particles of marine origin with the VLDR around 15%. The June 23$^{rd}$ 2019 dust outbreak consists primarily of high elevated dust concentrations, since $\delta_v$ values are greater than 15% (Fig. 9), after mid-day, with $\delta_p$ values reaching up to 30% in the height range of 3 to 5 km, which are representative of pure dust (Fig. 9). The dust plume was transported again directly from Sahara to Antikythera within 48 hours (Fig. 1d) and very low concentrations of dust particles are also present within the MABL.

## 4.2 Local mean E-field behaviour

Considering the electrical properties of the layers detected in Finokalia (Fig. 6 and Fig. 8), the LREF and the reconstructed mean electric field are depicted, with the local diurnal variation resembling the Carnegie curve. The $E_z$ values vary between a total minimum at ~ 05:00 (UTC) and the maximum at ~ 13:00 (UTC) with a mean value of ~ 173 V/m. An increase of the electric field is observed at about 22:00 (UTC) resulting in a double peak variations curve (Yaniv et al., 2016). The reconstructed mean E-field is close to the expected fair weather value and the slight difference can be attributed to local meteorological factors, atmospheric boundary layer characteristics (Anisimov et al., 2017) and the station's coastal location. Complementarily, $E_z$ diurnal variation in the station of Antikythera exhibits a minimum in early morning hours at ~ 23:00 (UTC) and a single maximum on early afternoon at ~ 19:00 (UTC) (Fig. 7 and Fig. 9), with a mean value of ~ 102 V/m. Since

the timeseries in Antikythera are restricted to one year, the mean E-field value is statistically biased, therefore it is lower than the expected fair weather value.

### 4.3 Observed E-Field enhancement as compared to LREF

In Fig. 6 and Fig. 7, we present the dust events that induced an enhanced electrical behaviour near the ground. The E-field strength measurements are averaged over 5 mins in order to be comparable with the lidar data. In the July 25th layer (Fig. 6),
dust advection is recorded since the first morning hours and areas of increased particle concentration can be spotted from early noon. The $\delta_p$ profile signifies that the layer consists primarily of dust which descends after ~16:00 (UTC) and falls entirely below 2 km at ~18:30 (UTC), but the mean electric field (black line) remains above the reference field (red contoured line), showing an increase when particle density is maximized towards noon and a small drop when dust concentrations within the MABL becomes significant.

A similar electrical behaviour was observed during the dust event of October 2018 that reached the PANGEA observatory. Large lofted particle concentrations are attributed to dust as discussed previously (Fig. 7). The mean $E_z$ appears enhanced as compared to the LREF, showing a further increase at ~ 21:00 (UTC) when dust deposition becomes prominent. According to the physical approximation of cylindrical charged areas (see Section 3.2), such an enhancement would be expected only when the lofted dust layer is electrically active and charge separation within the layer is prominent. From Fig. 5b, it becomes apparent
that the external E-field is more sensitive to charge imbalances, even small ones, than to separation distance variations, hence a charge imbalance within these layers could drive the E-field above the fair weather values, as observed in the above cases, for even smaller charge separation distances.

### 4.4 Observed E-Field reduction as compared to LREF

Several dust load cases were detected, both in Finokalia and Antikythera, where the near-ground electric field strength exhibits
a decrease when compared to the local reference field and, particularly, when high dust particle concentrations were present. In the specific study, we select the cases of March 2018 and June 2019 in terms of the similar temporal injection of dust particles, large AOD values and similar layer progression throughout the day (Fig. 8 and Fig. 9). From the $\delta_p$ profiles, we deduce that for both cases, the elevated layer between 2 and 4 km consists primarily of dust particles, while the decrease of $\delta_p$ towards the bottom of the layer is indicative of downward mixing inside the MABL, with marine particles of lower $\delta_p$s. The
mean E-field remains positive and well below the reference field, exhibiting an increase as dust injection initiates at ~09:00 UTC and then a decrease along the plume's progression (Fig. 8). The dust plume of June 2019 instils a similar electrical behaviour to the ground E-field, as the bottom of the layer seems to progressively move towards lower altitudes during late afternoon and the total dust load remains persistent. The mean E-field is positive and consistently below the reference field, exhibiting an increase close to fair weather values when particle injection begins towards noon and dust concentration is rising,
but later drops further below the LREF as the layer progresses to lower altitudes. Following the 1D model outputs for such a case (see Section 3.1.1), this observed reduction could be attributed to either electrically neutral dust aloft or to electrically

active dust with the charged regions in relatively small separation distances within the layer. Under the electrically active dust case, a charge imbalance of less than 10%, can be adequate to interpret the observed reduction of the E-field below the LREF for even smaller separation distances. But the detection of such an E-field reduction below the LREF cannot conclusively
characterize the electrical activity of the dust layer aloft.

### 4.5 Reversed E-field polarity

If a reversed polarity E-field is observed (in our timeseries there were dust cases under which the field exhibited polarity reversal), with the opposite sign signifying that the field vector points upwards instead of downwards, then the investigated formalism is capable of explaining the reversal. As such, a similar cylindrical configuration could be assumed with the only
450 difference being that the lower layer has to be negatively charged and the upper one, in the dipole case, to be positively charged. Under this condition, the conclusions derived from the model remain the same. Therefore, such an indication of reversal is explained only via reversed separated cylindrical charges and again points that lofted dust has to be electrified.

## 5 Discussion

### 5.1 E-field dependence on the bottom charged area height

From the above results, the question that arises is whether the proximity of the lower cylinder, to the ground itself, is capable to reproduce the electric field enhancement feature above the LREF. It becomes clear that two mechanisms act upon the enhancement of the ground electric field. The first is the decrease of the contribution of the upper layer as it moves upwards, due to the enhancement of the columnar resistance between the layer and the ground. The second is the increase of the contribution of the lower layer as it moves downwards, due to the decrease of the columnar resistance between the layer and
the ground. The closer the lower layer is to the ground the smaller the separation with the upper layer is required for the enhancement of the electric field.

In order to validate the influence of each parameter, we re-examine the ground E-field behaviour by keeping the lower cylinder at a fixed altitude of 2 km (close to the dust layer base, similarly to thundercloud activity e.g., Mallios and Pasko, 2012) and we, then, increase the separation distance. As observed in Fig. 10, the increasing separation distance causes the E-field to
465 increase at the ground and when it becomes large enough (top and bottom right panels), the upper cylinder does no longer influence the ground E-field. At this point, for both balanced and imbalanced dipoles with cylinder radius larger than ∼ 40 km, the field converges to a constant value. This becomes clearer when comparing Fig. 10 with Fig. 4. When the separation distance is 400 m, the electric field at the ground is larger than the reference field in the case of Fig. 10, while at Fig. 4, separation distance equal to 400 m happens when the bottom layer is at 2.75 km. In this case, the field is lower than the reference value
which indicates that the closer the bottom layer is to the ground, the smaller the separation distance needed for the enhancement of the ground electric field above the reference field. Moreover, the E-field value for zero separation distance is consistently

below the calculated fair weather value. As such, observations of enhanced E-field above the fair weather values, for dust driven days, can be reproduced only when an electrically active dust layer is transported above the field mill.

If we assume that the bottom charged area is close to the lofted layer base, we would expect an increase to the ground electric field as the layer progressively moves towards lower altitudes. For the comparison of the E-field timeseries with the descending layer base (Fig. 11), we use the cross component of the lidar attenuated backscatter coefficient at 532 nm, from which we can derive information on the vertical extent of the aerosol layers. More specifically, we applied a methodology where the first derivative of the attenuated backscatter coefficient is used to determine layer boundaries (Flamant et al., 1997; Mattis et al., 2008). The local maximum and local minimum of the derivative are considered to be the bottom and top of the layer respectively. The agreement between the height-time displays of the attenuated backscatter coefficient and the corresponding gradient (Fig. 6 to Fig. 9 and Fig. 11) can be used to verify the results of the gradient method.

As seen in the July 2017, March 2018 and June 2019 dust events, there is an enhancement of the reconstructed mean E-field followed by the layer base progression towards the ground, for specific timeframes within the day. This could signify the presence of positive charges accumulated to the layer base.

## 5.2 Chauvenet criterion validity

In Section 2.2.2, we described the processing chain for the determination of the local fair weather days at both atmospheric remote sensing stations. The novelty of the approach lies to the fact that only signal processing constraints are used, without incorporating criteria of local meteorological parameters that could redefine the initial conditions for the total fair weather days determination (Harrison and Nicoll, 2018). Nonetheless, threshold values concerning these factors are subjective, and may vary from study to study, which leads to differences in the extracted fair weather days. The specific study proposes a mathematically strict approach with the imposition of the Chauvenet criterion, which exploits only the field mill data and has a physical impact on the dataset. Under fair weather days, the mean electric field is approximately constant and the fewer by far dust driven days as captured in both stations, which are about 10% of the days within a typical year for both stations, will not influence significantly the reconstructed mean field value, but will be well beyond the standard deviation. The Chauvenet criterion excludes the days with such high variations as outliers and, therefore, the methodology for the reconstruction of the local reference field is less biased to variations occurring in dust driven days.

## 5.3 Generalization of the cylindrical model and LREF methodology

The methodology followed for the calculation of the ground electric field can be expanded to the area away from the central axis of the charged cylinders. As the cylinder radius increases and the infinite plate regime is approached, effects due to charged layer edges that induce radial electric field components, do not impact the sensor axis for a larger horizontal extent of the charged layer. This expands the analytical calculation as it becomes valid within a band region further away from the cylinder center. In the small radius regime, the sensor becomes sensitive to edge effects and the edge field can be far stronger than the on-axis field. If we assume that a transient dust layer is transported with a mean wind speed of 10 m/s, implying a regional

scale transport, then in a period of 2 hrs the edge will be 72 km away from the sensor axis (fast transits), a sufficient distance to not affect the vertical component of the electric field. Although these variations are present on the raw timeseries (observed peak activity in Fig. 6, Fig. 8 and Fig. 9), in the reconstruction of the LREF variations with timescales shorter than 6 hrs are the lower limit to the FFT input and are therefore excluded. This leaves the LREF unbiased to edge effects. Problems might be caused in our analysis in the case of very slowly moving dust layers, that are transported with wind speeds less than 1 m/s. Dust layer edge effects can provide basic information on the layer properties and could be incorporated in our cylindrical layer formalism, but this consists a subject of further investigation in the near future.

## 6 Summary and Conclusions

Near-ground electric field strength observations during Saharan dust advection over Greece exhibit three distinct responses of enhancement, reduction or sign reversal when compared to local fair weather values. In this paper, we present four cases of transient dust events that influence the ground electric field recorded at two atmospheric remote sensing stations synergistically with a lidar system and a field mill electrometer. Moreover, this work attempts to use only ground-based atmospheric electricity instrumentation as a proxy for electrified dust detection, with characterization in terms of optical properties from lidar observations. To quantify the effect of charged dust particles, we implemented a reference electric field representing the local fair weather field, using long-term measured timeseries, and examine the possible physical mechanisms that could explain the electric field behaviour. Our findings suggest that dust cases with the reconstructed mean E-field magnitude above the reference field indicate charge separation within the layer either as a balanced or imbalanced dipole (or a multipole) of charge layers, while when the mean field is completely below the reference field, dust electrical activity characterization is inconclusive. This ground electric field reduction below the local fair weather field can be attributed to either the conductivity reduction due to dust acting as a passive neutral element, where the greater the conductivity reduction the lesser the electric field reduction, or to charge separation between areas of accumulated charge.

The electrified dust scheme is approximated either via the absence of dust charge separation or with thin cylindrical finite charge geometries (as opposed to infinite plate analogues) that allow explaining the electric field dependence on the layer height and the separation distance between the regions of charge accumulation. Both concepts have been suggested to explain the observed E-field responses at ground. However, there is no observational evidence up to now to validate the charge strata morphology, which might be far from similar to the elevated layers morphology due to the charged dust particles complex transport dynamics. To constrain the modelling formalism proposed here, future research will include profiling of the columnar electrical properties of dust, deploying airborne platforms (balloons and UAVs) within the Saharan Air Layer during foreseen future experiments at Cyprus and Cape Verde within 2021.

**Appendix A**

**Dust layer acting as a passive element**

In Fig. A1, the dependence of the near ground electric field strength, $E_{z_0,layer}$ (red line) of an electrically neutral dust layer on the conductivity reduction factor, the scale height, the layer central height and the layer depth, as given in (11), is plotted and compared to the fair weather electric field $E_{z_0}$ at ground (blue line) which is given by (7). $E_{z_0}$ depends only on the scale height and decreases as $l$ increases, while it remains constant for the other varying parameters as expected from equation (7). The calculated fair weather field value of ~ 42 V/m, for the selected $l$, is comparable to the estimated value by Williams (2003)

from Ohm's law, when dividing the globally integrated conduction current density by the mean atmospheric electrical conductivity at ground ($J_{z_0} \approx 2x10^{-12} A/m^2, \sigma_0 \approx 5x10^{-14} S/m$) and assuming an exponentially increasing conductivity profile above the Earth's surface (Haldoupis et al., 2017). We note that this globally averaged value of $E_{z_0}$ is much less from the typically measured which is around 100 V/m (e.g., Corney et al., 2003; Reddell et al., 2004). We believe that the average value is more suitable for global calculations, because it incorporates the variations of the conductivity distribution around the

Earth. On the other hand, the typical value is tied to the location of the measurement, and it varies at different locations as the conductivity distribution changes. Consequently, $E_{z_0,layer}$ strongly depends on the conductivity reduction as depicted in the Fig. A1 case (a) curve, where the field reduces with the increasing reduction factor more effectively than with respect to the other three parameters, meaning that atmospheric conductivity reduction is the predominant factor that affects the E-field strength by largely lowering it. The $E_{z_0}$ depends only on the varying scaling height as expected from equation (7).

**Authors Contribution:** VD supervised the installation of the FM in both stations, collected and processed the data, constructed the reference field, provided physical input to the model and to the measurements' interpretation, plotted the lidar data and prepared the paper with contributions from all co-authors. SM conceptualized the model formalism and the processing chain of the reference field, also provided the key physical interpretation of the measurements. VA directed the preparation of the

paper, supervised the study, offered his specialty in lidar data interpretation and gave insight to the E-field measurements. JU kindly conferred the FM, co-supervised the study and provided scientific consultation on both model outputs and E-field measurements. GH installed the FM on both stations, ensured the continuous data retrieval and gave insight to the E-field measurements. AG operated the lidar and supervised the data retrieval, provided the processed lidar data, along with dust layer base plots and aided on their interpretation. IT provided the VLDR data and helped on the selection of the dust cases according

to these. Lastly, KT provided valuable scientific consultation concerning the E-field data, model assumptions and correlation to the dust layer proximity to the ground.

**Acknowledgments:** This research was supported by data and services obtained from the PANhellenic GEophysical Observatory of Antikythera (PANGEA) of NOA. The authors would like to acknowledge support of this work by the project

"PANhellenic infrastructure for Atmospheric Composition and climatE change" (MIS 5021516) which is implemented under

the Action "Reinforcement of the Research and Innovation Infrastructure", funded by the Operational Programme "Competitiveness, Entrepreneurship and Innovation" (NSRF 2014-2020) and co-financed by Greece and the European Union (European Regional Development Fund). We are grateful to EARLINET (https://www.earlinet.org/) and ACTRIS (https://www.actris.eu) for the data collection, calibration, processing and dissemination. VD would like to thank Dr. Eleni

Marinou for distributing the processing algorithm for the attenuated backscatter lidar retrievals. VD would also like to thank Prof. Charmandaris for his insightful comments and Dr. Nikos Kalivitis for his help in the data retrieval from Finokalia.

**Financial Support:** This research was supported by D-TECT (Grant Agreement 725698) funded by the European Research Council (ERC) under the European Union's Horizon 2020 research and innovation programme. VD would like to acknowledge

that this research is also co-financed by Greece and the European Union (European Social Fund- ESF) through the Operational Programme «Human Resources Development, Education and Lifelong Learning» in the context of the project "Strengthening Human Resources Research Potential via Doctorate Research" (MIS-5000432), implemented by the State Scholarships Foundation (IKY)» and supported by the A. G. Leventis Foundation scholarship. Support was provided also from the Stavros Niarchos Foundation (SNF) in the form of a student scholarship. KT acknowledges funding from the European Research

Council (ERC) under the European Unions Horizon 2020 research and innovation programme under grant agreement No. 771282.

**Conflicts of Interest:** The authors declare no conflict of interest.

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

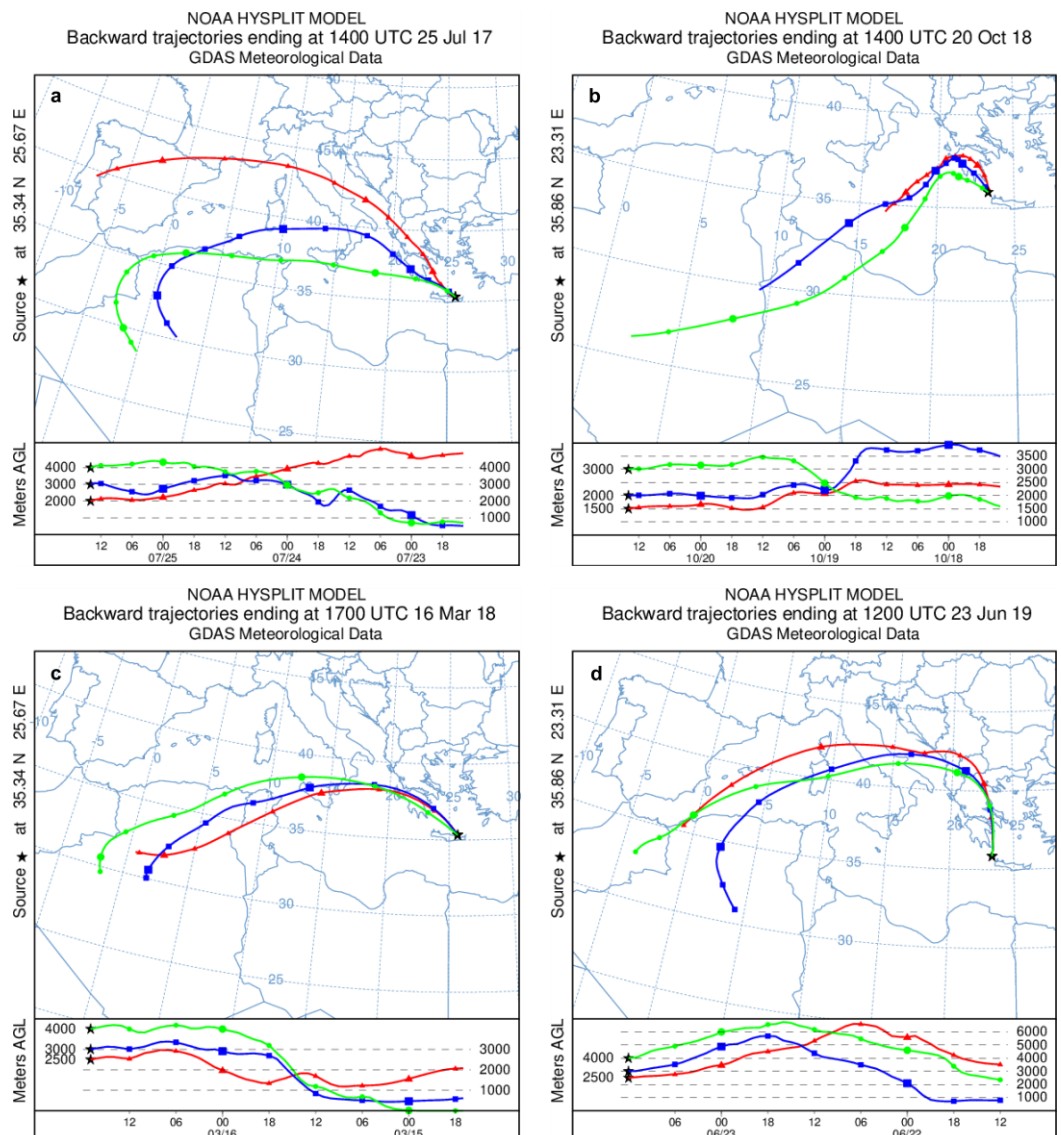

**Fig. 1.** NOAA HYSPLIT back trajectories model with GDAS assimilated data input for: **(a)** 25/07/2017 with 72 hrs backward propagation of air masses towards Finokalia, **(b)** 20/10/2018 with 72 hrs backward propagation of air masses towards Antikythera, **(c)** 16/03/2018 with 48 hrs backward propagation of air masses towards Finokalia and **(d)** 23/06/2019 with 48 hrs backward propagation of air masses towards Antikythera (https://www.ready.noaa.gov/HYSPLIT.php).

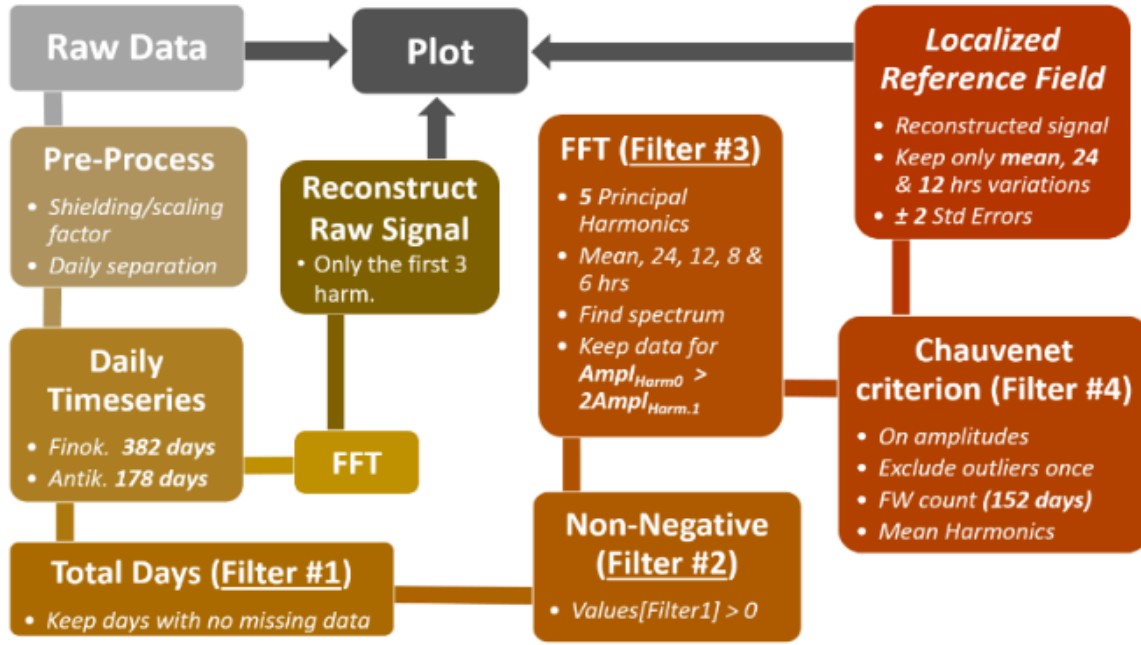

Fig. 2. Signal processing chain for: (i) the derivation of the Localized Reference Electric Field (LREF) that represents the local fair weather conditions, following the proposed filtering process (Filters no. 1 to no. 4, yellow-to-orange) of the field mill electric field data as retrieved on both Finokalia and Antikythera atmospheric stations and (ii) the derivation of the daily mean electric field under dust driven days from the same datasets (yellow-to-olive green). The LREF is compared to the mean electric field values in order to assess the electric field behaviour.

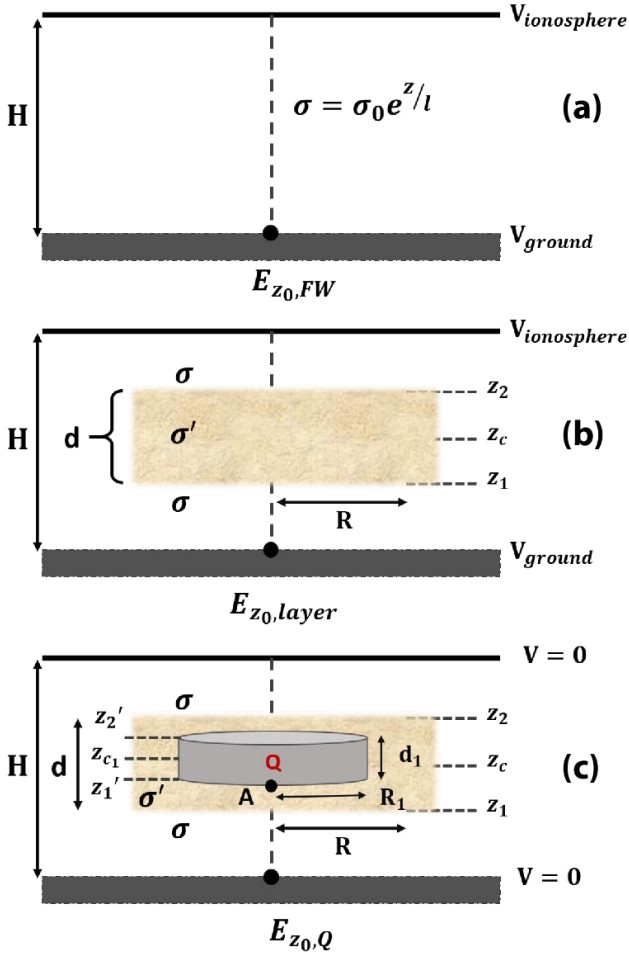

**Fig. 3. Schematic of the formalism for the calculation of the steady state near-ground electric field bounded between the ground potential ($V_{ground}$) and ionospheric potential ($V_{ionosphere}$) at height $H$, under: (a) fair weather conditions ($E_{z_0,FW}$) where atmospheric conductivity $\sigma$, follows an exponential distribution along the altitude $z$ with respect to the scaling height $l$, (b) the presence of an electrically neutral dust layer ($E_{z_0,layer}$) with depth $d$ and radius $R$, which modifies conductivity to $\sigma'$ and (c) the hypothesis of a cylindrical charged monopole ($E_{z_0,Q}$) within the dust layer, with depth $d_1$, radius $R_1$ and total charge $Q$. The monopole case is a superposition of the electrically neutral dust layer with the charged cylinder within the bounded atmospheric potential.**

**Table 1. Dust layer central height and depth, as derived from the PLDR profiles.**

| Dust Outbreak | $z_{c_i}$ $(km)$ | $d_i$ $(km)$ |
|---|---|---|
| **25/07/2017** (Fin.) | 3 | 4 |
| **20/10/2018** (Ant.) | 3 | 4 |
| **16/03/2018** (Fin.) | 3.5 | 2.5 |
| **23/06/2019** (Ant.) | 3.5 | 3 |

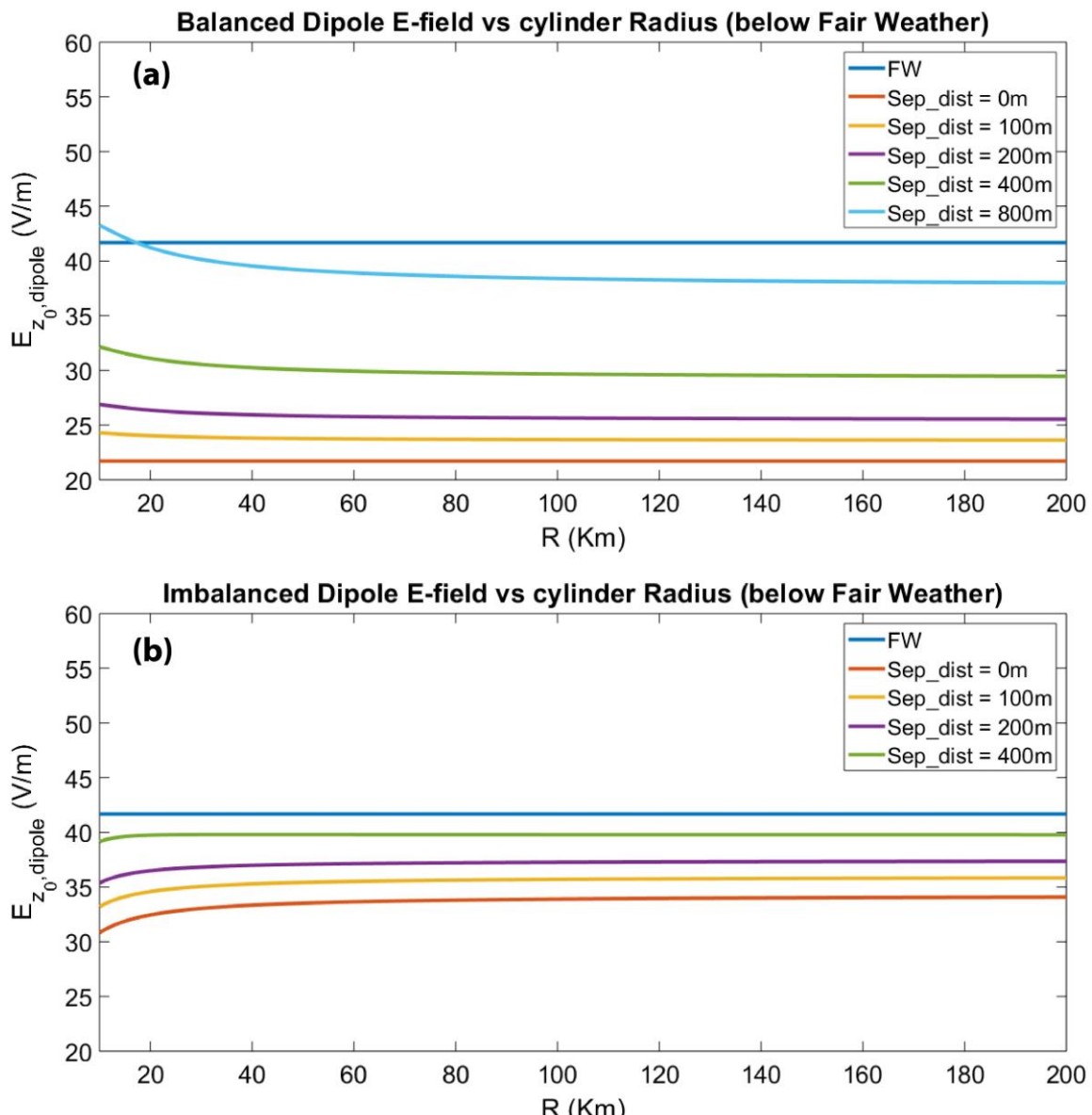

**Fig. 4.** Vertical electric field strength at ground level ($E_{z_0,dipole}$) below the fair weather field (blue line), for a dipole of: (a) finite uniformly charged cylinders and (b) non uniformly charged cylinders exhibiting a charge imbalance of 8%, within an elevated dust layer as a function of the cylinder radius $R$. $E_{z_0,dipole}$ is calculated for separation distances of 0 (electrically neutral dust), 100, 200, 400 and 800 m (balanced dipole case only) between the charged layers.

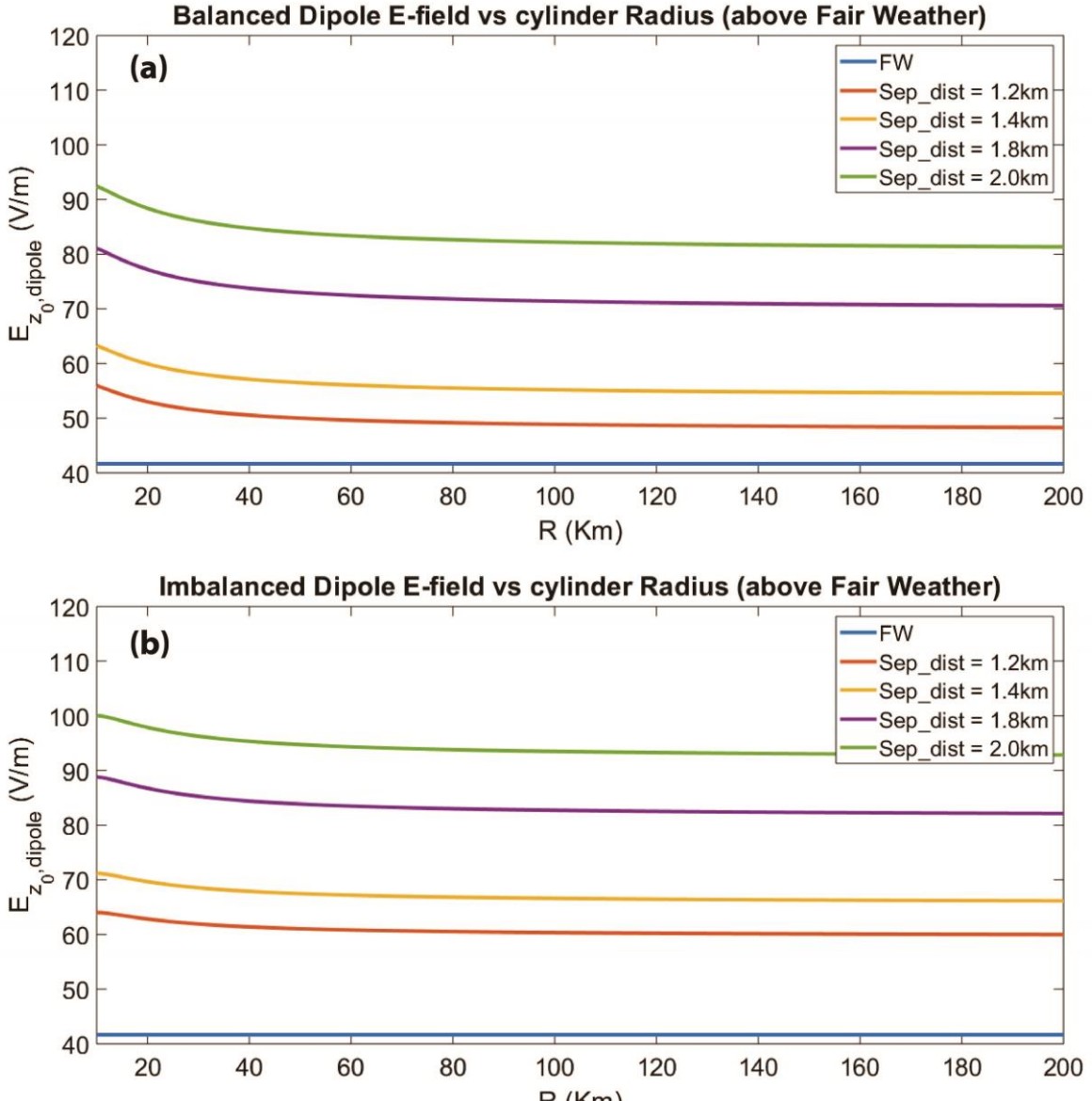

**Fig. 5.** Vertical electric field strength at ground level ($E_{z_0,dipole}$) above the fair weather field (blue line), for a dipole of: (a) finite uniformly charged cylinders and (b) non uniformly charged cylinders exhibiting a charge imbalance of 8%, within an elevated dust layer as a function of the cylinder radius $R$. $E_{z_0,dipole}$ is calculated for separation distances of 1.2, 1.4, 1.8 and 2 km between the two charged layers.

### Electric Field strength vs Attenuated Backscatter Coefficient at 532nm - July 25th 2017, Finokalia

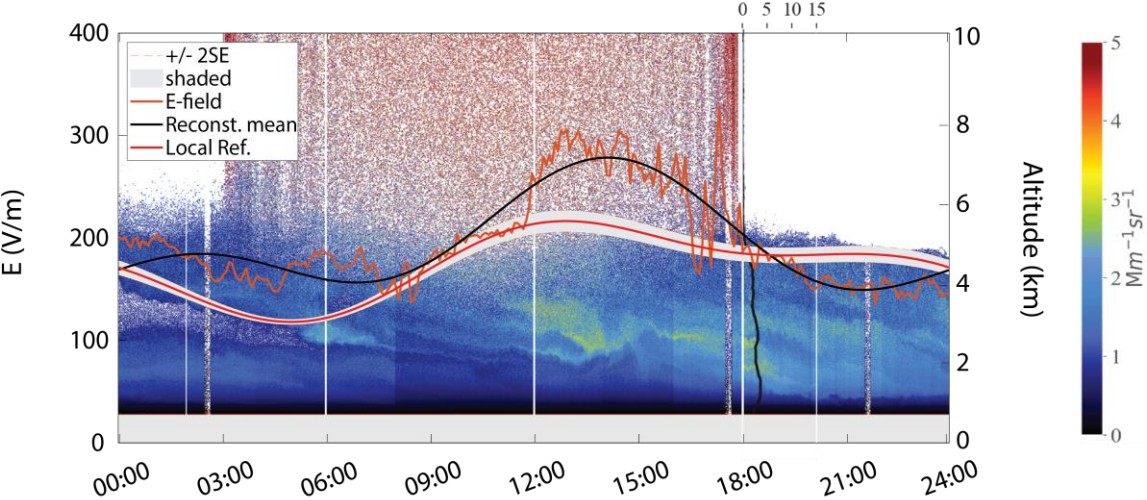

### Polly^XT Volume Linear Depolarization Ratio at 532nm - July 25th 2017, Finokalia

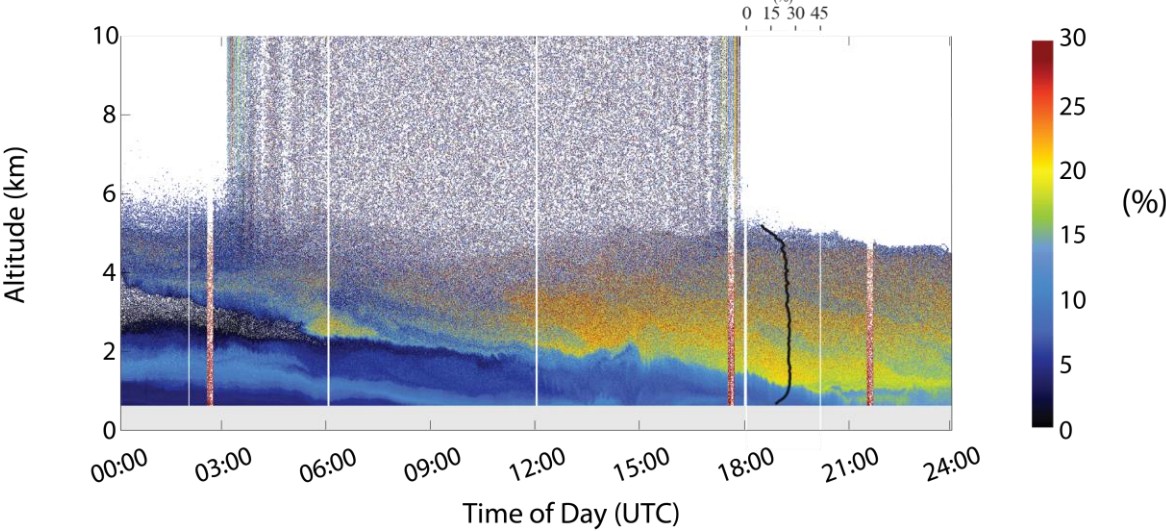

**Fig. 6.** Top panel**: Timeseries of the vertical electric field strength (orange), the extracted Localized Reference Electric Field (red) and the reconstructed mean electric field variation (black) from the field mill dataset, for the recorded 25/07/2017 dust layer in Finokalia. The E-field data are plotted with the time-height evolution of the attenuated backscatter coefficient (Mm$^{-1}$sr$^{-1}$) and the particle backscatter coefficient ($\beta$) profile (Mm$^{-1}$sr$^{-1}$, black vertical line) at 532 nm from the Polly$^{XT}$ lidar, which was co-located with the field mill. Areas of increased particle concentration are denoted with reddish tones, while the $\beta$ values are derived by averaging between 18:00 and 21:00 (UTC).** Bottom panel**: Volume Linear Depolarization Ratio ($\delta_v$, %) at 532 nm for the same dust layer as obtained from the Polly$^{XT}$ lidar and the Particle Linear Depolarization Ratio ($\delta_p$, %) profile (black vertical line) again averaged between 18:00 and 21:00 (UTC). Large $\delta_v$ values are depicted with reddish tones.**

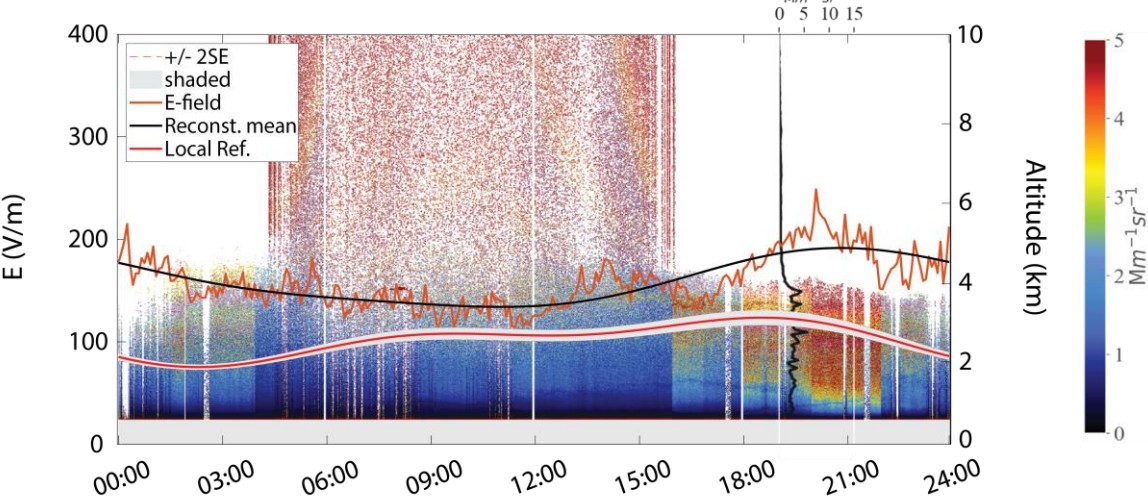

Electric Field strength vs Attenuated Backscatter Coefficient at 532nm - October 20th 2018, Antikythera

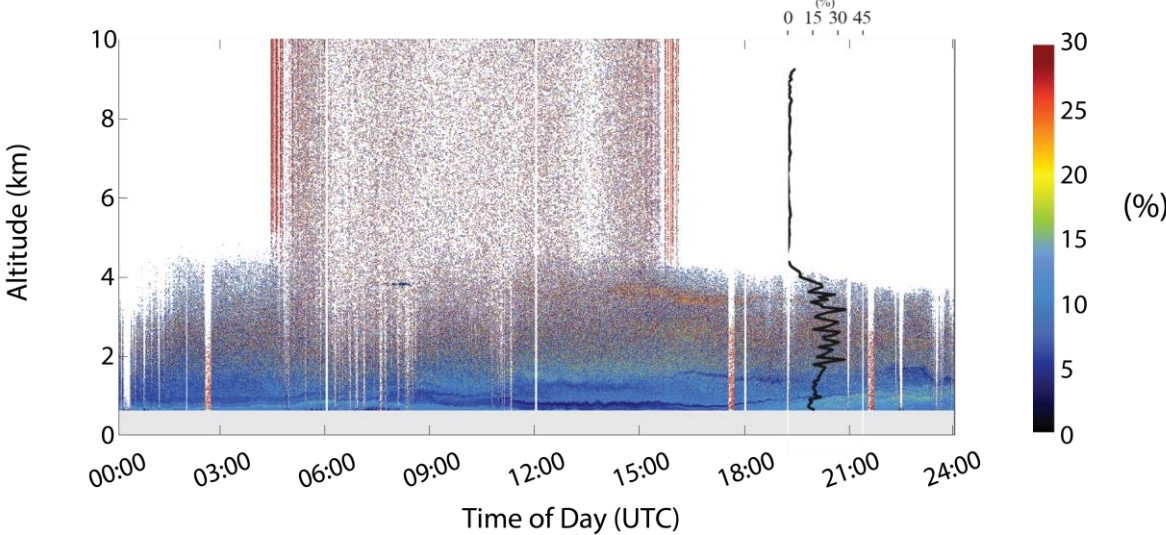

Polly^XT Volume Linear Depolarization Ratio at 532nm - October 20th 2018, Antikythera

**Fig. 7. Top panel: Timeseries of the vertical electric field strength (orange), the extracted Localized Reference Electric Field (red) and the reconstructed mean electric field variation (black) from the field mill dataset, for the recorded 20/10/2018 dust layer in Antikythera. The E-field data are plotted with the time-height evolution of the attenuated backscatter coefficient ($Mm^{-1}sr^{-1}$) and the particle backscatter coefficient ($\beta$) profile ($Mm^{-1}sr^{-1}$, black vertical line) at 532 nm from the Polly^XT lidar, which was co-located with the field mill. Areas of increased particle concentration are denoted with red tones, while the $\beta$ values are derived by averaging between 18:00 and 21:00 (UTC). Bottom panel: Volume Linear Depolarization Ratio ($\delta_v$, %) at 532 nm for the same dust layer as obtained from the Polly^XT lidar and the Particle Linear Depolarization Ratio ($\delta_p$, %) profile (black vertical line) again averaged between 18:00 and 21:00 (UTC). Large $\delta_v$ values are depicted with red tones.**

Electric Field strength vs Attenuated Backscatter Coefficient at 532nm - March 16th 2018, Finokalia

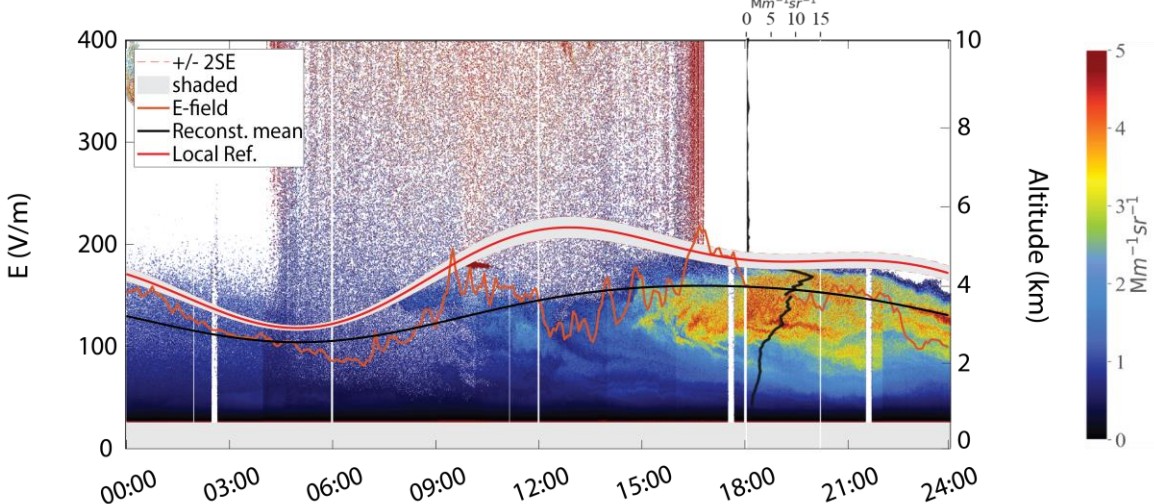

PollyXT Volume Linear Depolarization Ratio at 532nm - March 16th 2018, Finokalia

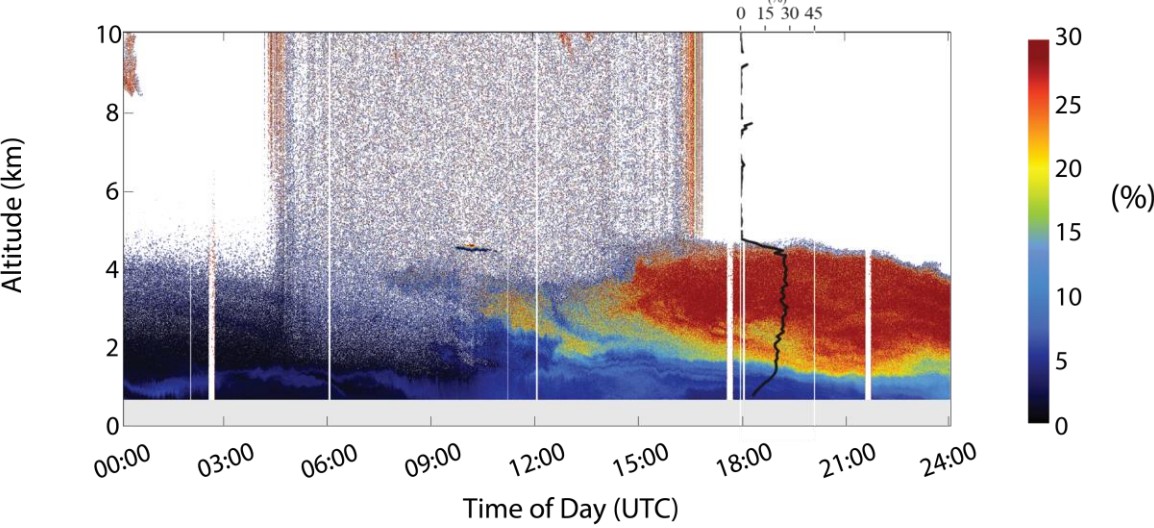

**Fig. 8. Top panel: Timeseries of the vertical electric field strength (orange), the extracted Localized Reference Electric Field (red) and the reconstructed mean electric field variation (black) from the field mill dataset, for the recorded 16/03/2018 dust layer in Finokalia. The E-field data are plotted with the time-height evolution of the attenuated backscatter coefficient ($Mm^{-1}sr^{-1}$) and the particle backscatter coefficient ($\beta$) profile ($Mm^{-1}sr^{-1}$, black vertical line) at 532 nm from the Polly$^{XT}$ lidar, which was co-located with the field mill. Areas of increased particle concentration are denoted with red tones, while the $\beta$ values are derived by averaging**
**between 18:00 and 21:00 (UTC). Bottom panel: Volume Linear Depolarization Ratio ($\delta_v$, %) at 532 nm for the same dust layer as obtained from the Polly$^{XT}$ lidar and the Particle Linear Depolarization Ratio ($\delta_p$, %) profile (black vertical line) again averaged between 18:00 and 21:00 (UTC). Large $\delta_v$ values are depicted with red tones.**

## Electric Field strength vs Attenuated Backscatter Coefficient at 532nm - June 23rd 2019, Antikythera

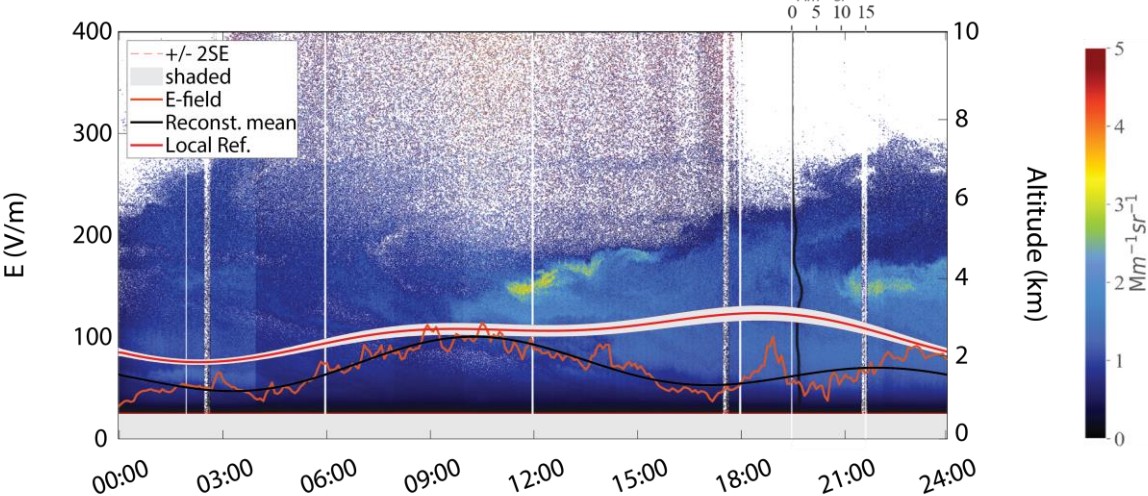

## Polly^XT Volume Linear Depolarization Ratio at 532nm - June 23rd 2019, Antikythera

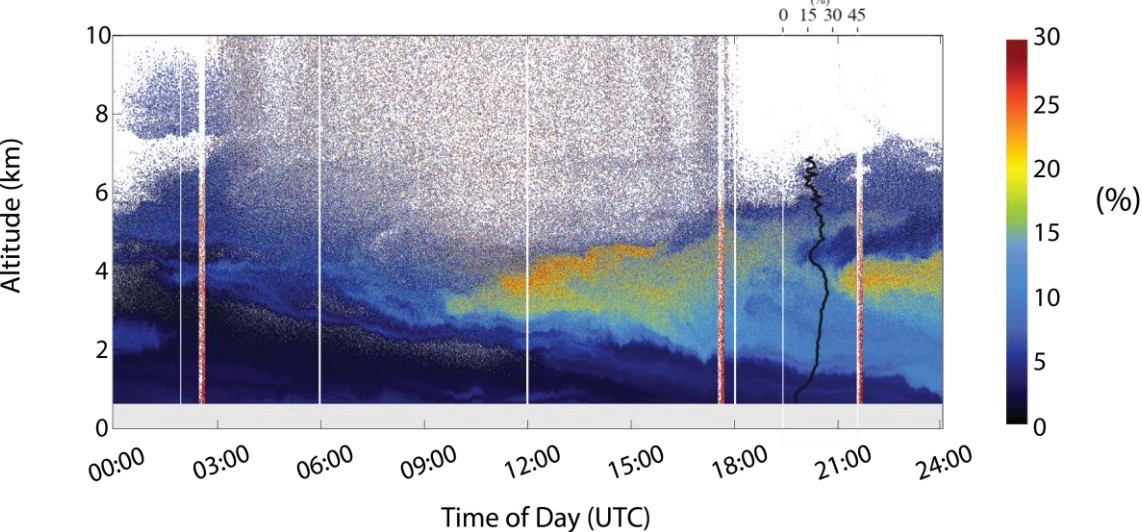

Time of Day (UTC)

**Fig. 9. Top panel:** Timeseries of the vertical electric field strength (orange), the extracted Localized Reference Electric Field (red) and the reconstructed mean electric field variation (black) from the field mill dataset, for the recorded 23/06/2019 dust layer in Antikythera. The E-field data are plotted with the time-height evolution of the attenuated backscatter coefficient (Mm⁻¹sr⁻¹) and the particle backscatter coefficient ($\beta$) profile (Mm⁻¹sr⁻¹, black vertical line) at 532 nm from the Polly^XT lidar, which was co-located with

835 the field mill. Areas of increased particle concentration are denoted with reddish tones, while the $\beta$ values are derived by averaging between 18:00 and 21:00 (UTC). Bottom panel: Volume Linear Depolarization Ratio ($\delta_v$, %) at 532 nm for the same dust layer as obtained from the Polly^XT lidar and the Particle Linear Depolarization Ratio ($\delta_p$, %) profile (black vertical line) again averaged between 18:00 and 21:00 (UTC). Large $\delta_v$ values are depicted with reddish tones.

Fig. 10. Dipole electric field strength at ground level ($E_{z_0,dipole}$) as a function of the cylinder radius $R$, with the bottom cylinder at 2 km fixed central height within the dust layer. The separation distance between the upper and bottom charged layer increases as the upper cylinder moves towards the top of the dust layer, for both cases of balanced and imbalanced dipoles.

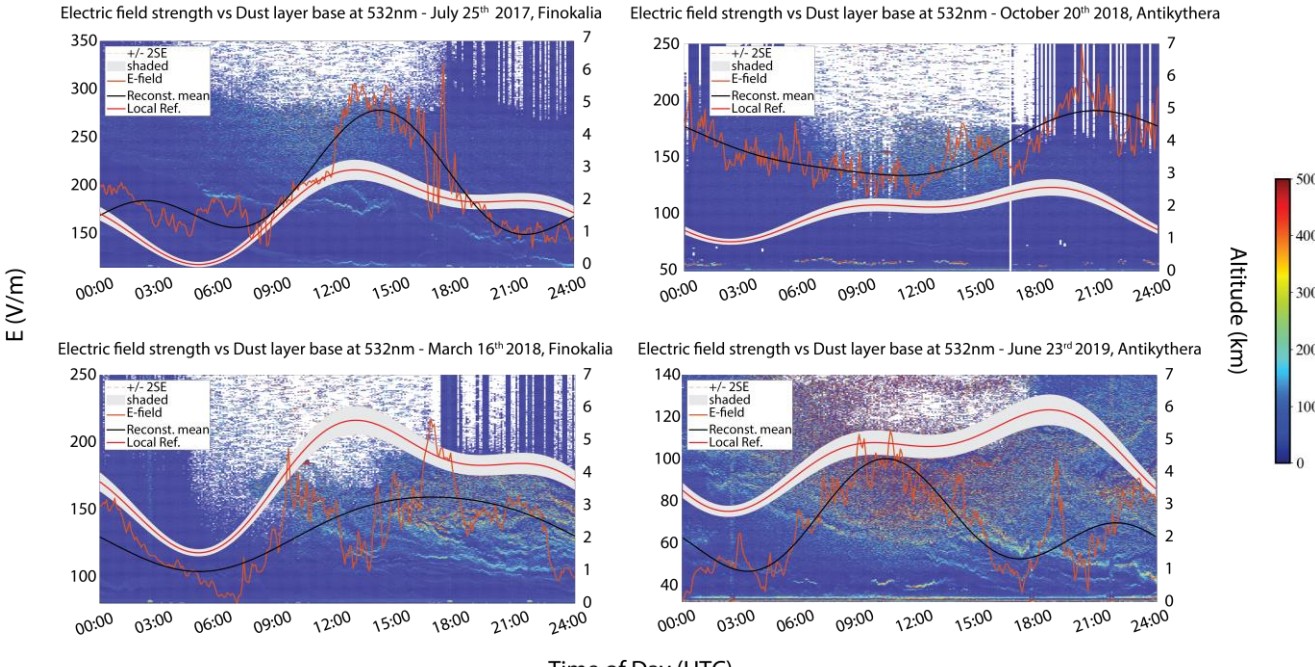

**Fig. 11. Timeseries of the vertical electric field strength (orange), the extracted Localized Reference Electric Field (red) and the reconstructed mean field variation (black), plotted with the first derivative of the cross component of the Polly[XT] attenuated backscatter coefficient at 532 nm against the altitude, for the dust cases of 25/07/2017, 20/10/2018, 16/04/2018 and 23/06/2019. The dust layer bottom base is signified by the positive maximum values of the derivative within the 0 - 500 colorbar range.**

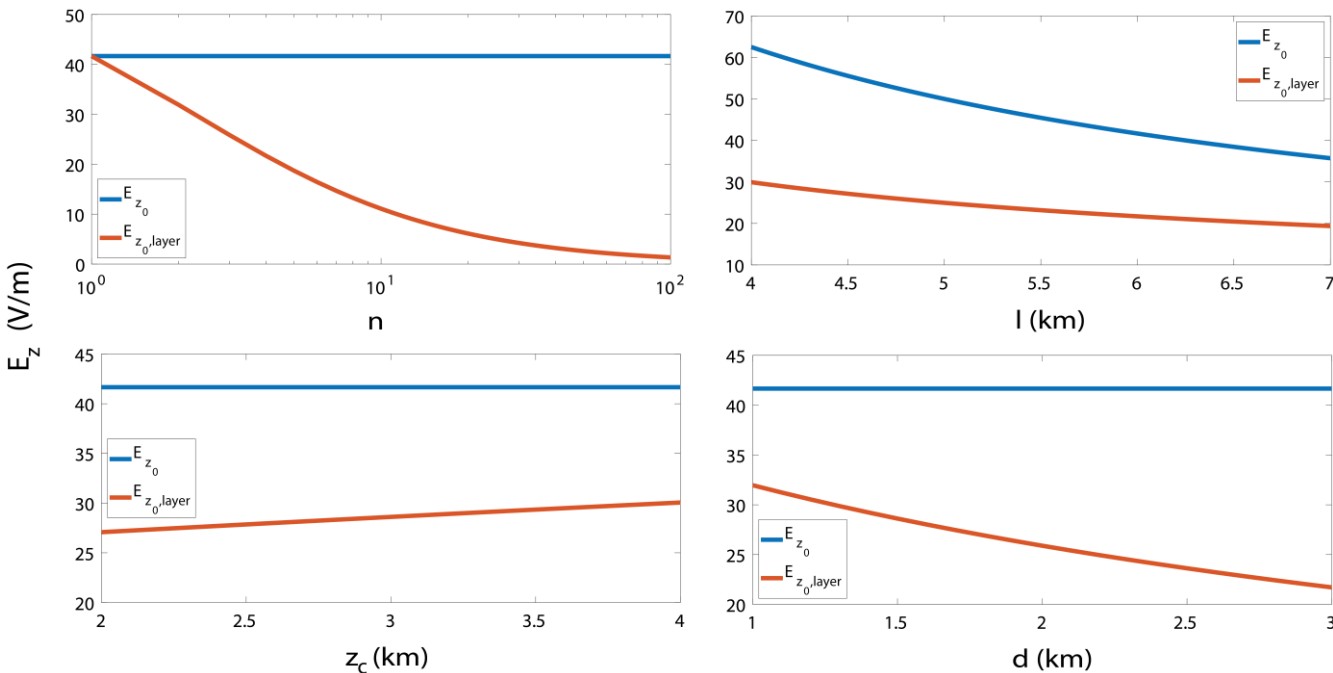

**Fig. A1. Dependence of the vertical electric field at ground level, under fair weather ($E_{z_0}$, blue line) and under the influence of an uncharged dust layer ($E_{z_0,layer}$, red line) on: (a) the reduction factor $n$, (b) the scaling height $l$, (c) the central layer height $z_c$ and (d) the dust layer depth $d$, for $1/\sigma_0 = 3 \cdot 10^{13} \ \Omega \ m$, $V_{ion} = 250$ kV and H = 70 km.**
