# Peer review of "The Electrical Activity of Saharan Dust as perceived from Surface Electric Field Observations"

_Atmospheric Chemistry and Physics, 2020_

## Referee Comment (RC1) · Konstantinos Kourtidis (Referee) · 19 Aug 2020

Major comments:

1.The title does not exactly reflect the contents of the manuscript. The manuscript, apart from the surface E-field observations mentioned in the title, also presents height profiles of dust AND e-field modelling. "The electrical activity of Saharan dust" suits better the contents to my taste.

2. The different Y-axes in Figs. 4-7 of the E-field, make it extremely difficult for (or even deter) the reader to compare the observations.

[Figure]

Minor comments:

Lines 37-39: The statement that "The Global Electric Circuit (GEC) is an electrical circuit, and specifically a spherical capacitor that is formed between two conducting planes, with one being the Earth's surface, a good conductor of electricity, and the other the Ionosphere" is not correct. The GEC CAN BE THUS COCEPTUALISED but it IS NOT what it is stated. Please rephrase.

Line 40: "GEC, greatly depend on ambient weather conditions and convective meteorological systems...", a passing reference here would also be Kourtidis et al., The influence of circulation weather types on the exposure of the biosphere to atmospheric electric fields, International Journal of Biometeorology DOI: 10.1007/s00484-020-01923-y

Lines 85-86 "In this study, we focus on monitoring perturbations of the E-field near the ground caused by the transported dust layers, with special emphasis on slow E-field perturbations (with period larger than 6 hours), ...": Period or duration?

Lines 91-92: "by the ground-based electrometer " → "by ground-based electrometers".

Lines 106 "are prevalent during the intermediate season ": which season is that? Please clarify.

Line 128-129 "VLDR is defined as the ratio of the cross–polarized to the parallel-polarized backscattered signal . . . and typical pure dust values are between 30% - 40% ...": Please give some information also on VLDR values for mixed dust as well as VLDR values above 40%.

Line 155, for LREF: "The specific reference field represents the electric field behavior under local fair weather conditions, ...", but at line 158 "the local fair weather days are classified as the less electrically disturbed days, ...". So, does LREF represent fair weather days or not so unfair weather days? Also, it is not so clear to me what do the authors mean by fair weather. I am unclear about wheather they mean days representative of GEC influence.

Line 168-169 "first five principal harmonics to the diurnal cycle of the electric field ...": Please add something to help the reader understand why the diurnal cycle of the field should or could represented by 5 harmonics.

Lines 174-175 "fi = i * (t/24) * 360 is the frequency of each harmonic ": is t the same as in eq. (1) above? If yes, then fi is a time-varying frequency? I believe t should be removed, else it results in terms of t^2 in (1). I believe the correct is fi = (i/24) * 360.

Line 204 "Under fair weather atmospheric conditions, complete lack of particles in the atmospheric circulation is expected ": Not by me. By whom is COMPLETE LACK of particles expected? I believe the sentence should be rephrased to a less emphatic form.

Line 209 "near ground atmospheric conductivity and the atmospheric scale height...": So, you assume $\sigma$ to be proportional to density I guess. Perhaps you could add a line or two on the foundations of this assumption.

Line 246: A reference to Instructor's Solution Manual for Introduction to Electrodynamics, 4th Edition, 2013, is needed.

Figure 1 could (and should, to my taste) be incorporated into Figs 4-7.

Figs 4 and 5 captions "VLDR values between 25% and 30% indicate the presence of mostly mixed dust ", Figs 6 and 7 captions "VLDR values between 35% and 40% indicate a pure dust layer ": Values >30% but <35% indicating what?

---

## Referee Comment (RC2) · Anonymous Referee #2 · 23 Aug 2020

The authors study four dust events in Greece, using a combination of ground-based electric field measurements and lidar. Âă The events involve dust that originated in the Sahara 48 to 72 hours previously. Âă Two of the events enhance the electric field relative to the reference fair weather field, and the other two events diminish the electric field.

This paper presents a simple model to describe these electrical effects. Âă There are two components of the model. Âă First, that the dust will reduce the conductivity in the region it occupies by scavenging ions; this effect occurs even with neutral dust particles. Âă Second, there could be regions of charged dust – this is modeled as cylinders of monopolar charge (there could be two cylinders, one of positive and one of negative

charge). Some of the parameters for the model can be obtained from the lidar, while other parameters cannot be independently obtained.

Here is where I get lost.Âǎ I found the results section very hard to follow.Âǎ It appears to me the authors show experimental results for dust event (Figs 4-7), and then present results of the model under various parameters (Figs 8-13).

I do not think there is much interest in the results of the simple model under various parameters.Âǎ I think these figures and the associated text should be removed.

Rather, I think they should focus (succinctly) on using the model to rationalize the experimental results.Âǎ This must be done much better in order for the paper to be publishable

Also, it is important to justify the assumptions in the model (this is much more important than the mathematical details, which they cover in great depth). – give physical reason why eqn 4 has this form – why do uncharged aerosol particles scavenge ions?Âǎ This is a key assumption for their model, as it leads to the reduction factor n, but its not clear to me that this is physically correct. The authors must provide strong evidence to support this

And overall, I think the paper needs to be communicated much more clearly, and walk the reader through the results and the logic behind their ideas.Âǎ Figure captions should clarify what the data represents (cannot assume someone knows this).Âǎ As I said above, I got lost and couldn't understand things.

---

## Author Comment (AC1) · 24 Nov 2020

**RESPONSE TO REFEREE#1 COMMENTS AND PEER-REVIEW REPORT**

Manuscript Title: **"The Electrical Activity of Saharan Dust as perceived from Surface Electric Field Observations in Greece"**

Authors (as declared in the submitted manuscript with an addition of Ms. Ioanna Tsikoudi, justified contribution to the *declaration of author contributions*):

**Vasiliki Daskalopoulou, Sotirios A. Mallios, Zbigniew Ulanowski, George Hloupis, Anna Gialitaki, Ioanna Tsikoudi, Konstantinos Tassis and Vassilis Amiridis**

Dear respected Editor/Reviewers,

The authors highly appreciate the comprehensive feedback throughout the review process and kindly integrate the reviewer comments in the revised preprint manuscript, as follows:

**REFEREE#1 COMMENTS:**

*"Major comments:*
*1. The title does not exactly reflect the contents of the manuscript. The manuscript, apart from the surface E-field observations mentioned in the title, also presents height profiles of dust AND e-field modelling. "The electrical activity of Saharan dust" suits better the contents to my taste."*

**Author's Response:** We highly appreciate the insightful comment on the paper title, which indeed would describe the context in a broader manner, but in our effort to highlight the synergistic measurements of the ground E-field and particle optical properties, but also point out that the attempt for a distinction of electrically active dust is done only by surface E-field observations, the authors would like to keep the title as is. We have, however, excluded the geographical information which we find limiting of our methodology. Therefore:

Title from: **"The Electrical Activity of Saharan Dust as perceived from Surface Electric Field Observations in Greece"** is changed to **"The Electrical Activity of Saharan Dust as perceived from Surface Electric Field Observations"**

*2. The different Y-axes in Figs. 4-7 of the E-field, make it extremely difficult for (or even deter) the reader to compare the observations."*

**Author's Response:** Figures 4 to 7 (revised as Fig. 6 to Fig. 9) have been adequately changed to the same Y-axis limits. Multi-panel Figure 14 (revised to Fig. 11) has been kept with the max E-field values for each dust event in order to: i. fully exploit the min/max range of the measured field for each event, ii. to highlight the comparison between the derived mean E-field (black line) with the reference field (red line), which is one of the key features of the study and lastly, iii. to enhance the visual comparison between the E-field diurnal variation and the dust layer bottom base throughout the day.

*"Minor comments:*
*Lines 37-39: The statement that "The Global Electric Circuit (GEC) is an electrical circuit, and specifically a spherical capacitor that is formed between two conducting planes, with one being the Earth's surface, a good conductor of electricity, and the other the Ionosphere" is not correct. The GEC CAN BE THUS COCEPTUALISED but it IS NOT what it is stated. Please rephrase."*

**Author's Response:** Thank you again for the comment, we have modified **Lines 38-42** as:

"The Global Electric Circuit (GEC) represents the electric current pathway in the Earth's atmosphere. The electric current that flows upwards from thunderstorms and electrified clouds into the Ionosphere, spreads out over the globe along magnetic field lines to the opposite hemisphere, and returns to the surface of the Earth as the fair weather air-Earth current (Bering et al, 1998). The GEC is established by the conducting atmosphere sandwiched between the conductive Earth and the conductive Mesosphere/Ionosphere (Williams, 2009)."

**References** (*not previously included in the manuscript*):

Bering III, E.A., Few, A.A., Benbrook, J.R., The global electric circuit, Phys. Today, 51, 24–30, 1998.

Williams, E.R., The global electric circuit: A review, Atmos. Res., 91, doi:10.1016/j.atmosres.2008.05.018, 2009.

*Line 40: "GEC, greatly depend on ambient weather conditions and convective meteorological systems...", a passing reference here would also be Kourtidis et al., The influence of circulation weather types on the exposure of the biosphere to atmospheric electric fields, International Journal of Biometeorology DOI: 10.1007/s00484-020-01923-y*

**Line 40**, modified to **Line 44:** added the respective reference as (Kourtidis et. al, 2020), thank you for the kind suggestion.

**References** (*not previously included in the manuscript*):

Kourtidis, K., Szabóné André, K., Karagioras, A., Nita, I. A., Sátori, G., Bór, J. and Kastelis, N.: The influence of circulation weather types on the exposure of the biosphere to atmospheric electric fields, Int. J. Biometeorol., doi:10.1007/s00484-020-01923-y, 2020.

*Lines 85-86 "In this study, we focus on monitoring perturbations of the E-field near the ground caused by the transported dust layers, with special emphasis on slow E-field perturbations (with period larger than 6 hours), ...": Period or duration?*

**Lines 90:** we have changed the word "period" to "duration".

*Lines 91-92: "by the ground-based electrometer " ! "by ground-based electrometers".*

**Lines 91-92** updated to **Lines 95-96:** added the word "same" as "by the same ground-based electrometer", as it was the same instrument that we re-located from Finokalia to Antikythera station.

*Lines 106 "are prevalent during the intermediate season ": which season is that? Please clarify.*

**Line 106** updated to **Line 111:** added the phrase "of March till June" as in "are prevalent during the intermediate season of March till June"

*Line 128-129 "VLDR is defined as the ratio of the cross–polarized to the parallel-polarized backscattered signal…and typical pure dust values are between 30% - 40% …": Please give some information also on VLDR values for mixed dust as well as VLDR values above 40%.*

**Author's Response:** The authors have re-evaluated the Volume Linear Depolarization Ratio (VLDR) range of values, as the ones used in the pre-print version were characteristic of the Particle Linear Depolarization Ratio (PLDR) used in earlier stages of this research. Moreover, during the review process of the manuscript, the Polly[XT] lidar underwent

a refined polarization calibration process that altered the values used for the derivation of the PLDR and the data were, thereafter, processed uniformly with the Single Calculus Chain (SCC) algorithm (D'Amico et al., 2015) of EARLINET. Therefore, we have incorporated the novel plots to the final version of the manuscript and in order to further strengthen dust layer characterization, we have calculated profiles of the particle backscatter coefficient (β) and the PLDR (referred to as $\delta_p$ in the revised manuscript), averaged for timeframe when dust episodes were fully developed. We, have therefore carefully inspected the manuscript and corrected the aerosol subtypes within the following ranges (Veselovskii, 2016, 2020).

- PLDR values for pure Saharan dust: 25 – 35% at 532nm
- VLDR values above 15% at 532nm, are indicative of dust presence (i.e. mixtures of dust with weakly depolarizing aerosols such as marine or smoke particles)

**Renewed Plot example:**

Electric Field strength vs Attenuated Backscatter Coefficient at 532nm - March 16th 2018, Finokalia

[Figure]

PollyXT Volume Linear Depolarization Ratio at 532nm - March 16th 2018, Finokalia

[Figure]

**Fig. 8.** **Top panel:** Timeseries of the vertical electric field strength (orange), the Localized Reference Electric Field (red) and the reconstructed mean electric field variation (black) plotted with the time-height evolution of the attenuated backscatter coefficient (Mm⁻¹sr⁻¹) and the particle backscatter coefficient (β) profile (Mm⁻¹sr⁻¹, black vertical line) averaged between 18:00 and 21:00 (UTC), for the 16/03/2018 dust layer in Finokalia station. Areas of increased particle concentration are denoted with red tones, while the β values reach up to 15 (Mm⁻¹sr⁻¹). The mean E-field remains positive but well below the reference field, exhibiting an increase as particle injection initiates at ~11:00 (UTC) and then a decrease along the plume's progression. **Bottom panel:** Volume Linear Depolarization Ratio (VLDR, %) for the same dust layer as obtained from the PollyXT lidar and the Particle Linear Depolarization Ratio (PLDR, %) profile (black vertical line). VLDR values close to 30% are indicative of high dust particle concentration and PLDR values persistently of 30% are characteristic of pure dust within the entirety of the layer (1-4 km).

All related paragraphs and figure captions within the text have been updated respectively. More specifically:

- **Lines 124-129:** modified to Paragraph from **Line 129** to **Line 142** as "The system employs two detectors, … of substantial particle concentrations (Haarig et al., 2017; Veselovskii et al., 2016, 2020)."
- **Lines 262-270:** modified to Paragraph from **Line 370** to **Line 388** as "The July 2017 and March 2018 dust events…, are also present within the MABL."
- **Fig. 4** (renewed Fig. 6) caption in **Lines 788-795**: "Top panel: Timeseries of the vertical electric field… δp values between 25% - 30% in the afternoon are characteristic of pure dust."
- **Fig. 5** (renewed Fig. 7) caption in **Lines 799-807**: "Top panel: Timeseries of the vertical electric field… δp values between 25% - 30% in the afternoon are characteristic of pure dust."
- **Fig. 6** (renewed Fig. 8) caption in **Lines 811-819**: "Top panel: Timeseries of the vertical electric field… PLDR values persistently of 30% are characteristic of pure dust within the entirety of the layer (1-4 km)."
- **Fig. 7** (renewed Fig. 9) caption in **Lines 823-831**: "Top panel: Timeseries of the vertical electric field… δp values between 25% - 30% in the afternoon are characteristic of pure dust."

**Reference:**

D'Amico, G., Amodeo, A., Baars, H., Binietoglou, I., Freudenthaler, V., Mattis, I., Wandinger, U., and Pappalardo, G.: EARLINET Single Calculus Chain – overview on methodology and strategy, Atmos. Meas. Tech., 8, 4891–4916, https://doi.org/10.5194/amt-8-4891-2015, 2015.

**References** (*not previously included in the manuscript*):

Veselovskii, I., Goloub, P., Podvin, T., Bovchaliuk, V., Derimian, Y., Augustin, P., Fourmentin, M., Tanre, D., Korenskiy, M., Whiteman, D. N., Diallo, A., Ndiaye, T., Kolgotin, A. and Dubovik, O.: Retrieval of optical and physical properties of African dust from multiwavelength Raman lidar measurements during the SHADOW campaign in Senegal, Atmos. Chem. Phys., 16(11), 7013–7028, doi:10.5194/acp-16-7013-2016, 2016.

Veselovskii, I., Hu, Q., Goloub, P., Podvin, T., Korenskiy, M., Derimian, Y., Legrand, M. and Castellanos, P.: Variability in lidar-derived particle properties over West Africa due to changes in absorption: Towards an understanding, Atmos. Chem. Phys., 20(11), 6563–6581, doi:10.5194/acp-20-6563-2020, 2020.

*Line 155, for LREF: "The specific reference field represents the electric field behavior under local fair weather conditions, ...", but at line 158 "the local fair weather days are classified as the less electrically disturbed days, ...". So, does LREF represent fair weather days or not so unfair weather days? Also, it is not so clear to me what do the authors mean by fair weather. I am unclear about whether they mean days representative of GEC influence.*

**Authors' Response:** The Carnegie Vessel cruises retrieved the first ship-borne measurements of the PG which were processed by the Carnegie Institute researchers back in the early 20th century. The lack of consistent meteorological parameters monitoring along the cruises could not enable the characterization of fair weather days as what was later established by UK Met Office as "fair weather conditions" (Harrison, 2013). Therefore, fair weather days were selected as the representative days of the least electrically disturbed conditions, which is what we adopt also in the present work as fair weather. The proposed Reference field, as it omits fast perturbations (either from transient convective systems or extreme meteorological conditions that are translated to extreme E-field variabilities), represents the fitting to the local GEC influence without incorporating meteorological parameters (Shatalina, 2019). The subsequent necessity that arises for further validation of LREF through the comparison to the fair weather field constrained by distinct meteorological criteria, similar to what is discussed in the work of Harrison and Nicoll (2018), is part of ongoing work undertaken by the authors.

Hence, the term of local fair weather days would be more clearly communicated by the term electrically fair weather days. In order to communicate the distinction more clearly, the authors have revised the following paragraph as suggested:

**Modified Paragraph in Lines 152-161:** From "For the classification of the behaviour of the vertical electric field under dust influenced conditions, as that of an enhanced, reduced or reversed E-field, … are clearly dominated by local influences (Harrison & Nicoll, 2018)." to:

**Lines 165-175:** "The classification of the vertical electric field behavior under dust influenced conditions, as that of an enhanced, reduced or reversed E-field, necessitates comparison with the local long-term fair weather electric field. In order to represent solely the diurnal GEC influence at each observational site, away from electric generators perturbing the near ground E-field (e.g. Zhou and Tinsley, 2007), we construct a Localized Reference Electric Field (LREF) by exploiting only the timeseries inherent attributes and the measuring quantity itself, through the processing chain described below (Fig. 2). Various authors have presented different methodologies for determining fair weather conditions (e.g. Anisimov, 2014). For the specific study, the selected constraints of fair weather are based on the classification of fair weather days as the less electrically disturbed days, also assumed by the Carnegie Institute researchers (Harrison, 2013). Although, local effects on the E-field at each site can be of random nature (wind gusts, lightning strikes, radon emission and turbulent flows due to orography), the selection of fair weather data can be based on noise reduction by subtracting values which are clearly dominated by local influences and not directly addressing the meteorological criteria of fair weather (Harrison & Nicoll, 2018)."

**References** (*not previously included in the manuscript*):

Shatalina, M. V., Mareev, E. A., Klimenko, V. V., Kuterin, F. A. and Nicoll, K. A.: Experimental Study of Diurnal and Seasonal Variations in the Atmospheric Electric Field, Radiophys. Quantum Electron., 62(3), 183–191, doi:10.1007/s11141-019-09966-x, 2019.

Zhou, L. and Tinsley, B. A.: Production of space charge at the boundaries of layer clouds, J. Geophys. Res. Atmos., 112(11), 1–17, doi:10.1029/2006JD007998, 2007.

*Line 168-169 "first five principal harmonics to the diurnal cycle of the electric field ...": Please add something to help the reader understand why the diurnal cycle of the field should or could represented by 5 harmonics.*

**Author's Response:** added the following sentences and restructured paragraph **Lines 162-170** "The FM data are…. in the following signal equation for S(t):" to:

**Lines 179-189:** "When representing the E-field diurnal variation by the Carnegie Curve, which is used consistently as a reference against locally measured atmospheric electricity parameters, the hourly variations of the field that shape the curve correspond to the 24, 12, 8, and 6-hour durations, as deduced from previous consistent observations of the Carnegie vessel (Harrison, 2013). The present study attempts to derive the local harmonic fit in the form of LREF, based on the form of the Carnegie curve, and assuming that this trend should be followed by the reference field as well. Consequently, the averaged 1s data to 1-minute data (datalogger configuration) are shifted to the frequency domain through a Fast Fourier Transform (FFT) representation so as to evaluate the relative contributions of the first five principal harmonics to the diurnal cycle of the electric field (hourly variations including daily mean), which are depicted in the following signal equation for S(t) (1). We note, that days with missing data are removed, because the uneven temporal distribution of the measurements modifies the time window for the FFT algorithm, and therefore, modifies the timeseries spectrum."

*Lines 174-175 "$f_i = i * (t/24) * 360$ is the frequency of each harmonic ": is t the same as in eq. (1) above? If yes, then $f_i$ is a time-varying frequency? I believe t should be removed, else it results in terms of $t^2$ in (1). I believe the correct is $f_i = (i/24) * 360$.*

**Lines 174-175** updated to **Lines 189-191:** the presence of time t in Eq. (1) is a typographical error and therefore removed as follows:

$$S(t) = A_0 + A_1 cos(2\pi f_1 + \varphi_1) + A_2 cos(2\pi f_2 + \varphi_2) + A_3 cos(2\pi f_3 + \varphi_3) + A_4 cos(2\pi f_4 + \varphi_4)$$

where now the frequency $f_i$ relates exactly to the given term as $f_i = i \frac{t}{24} 360°, for \ i = 0, ... 4$.

**Line 171** updated to **Line 191:** changed sentence from "the electric field at time t in hrs" to "the electric field at time t in hrs (UTC)".

*Line 204 "Under fair weather atmospheric conditions, complete lack of particles in the atmospheric circulation is expected ": Not by me. By whom is COMPLETE LACK of particles expected? I believe the sentence should be rephrased to a less emphatic form.*

**Authors' Response:** We again thank the reviewer for the insightful comment, as we agree that under fair weather the electric field near the ground and in turn the air-Earth conduction current are strongly influenced by the variable concentration of aerosol particles (diurnal, annual and seasonal) (e.g. Siingh et al., 2009; Kastelis et al., 2016 and references therein), cloud droplets and water vapor, that in turn affect the ion attachment and recombination processes (e.g. Tinsley, 2006; Rycroft, 2008). Ideally, the complete lack of aerosols will lead to electrical properties (ion density, electric field) exactly the same as the unperturbed fair weather values. This means that the losses of the atmospheric ions are attributed only to the ion-ion recombination mechanism. As the concentration of the aerosols increases, perturbations from the fair weather values occur, since ion attachment to aerosols begins to become prominent. Nevertheless, as the reviewer states, there can be aerosol particles with concentrations that do not alter significantly the fair weather values.

**Line 204:** The sentence "Under fair weather atmospheric conditions, complete lack of particles in the atmospheric circulation is expected, except of ions." is replaced with updated:

**Lines 222-227**: "Ideally, under strict fair weather conditions, complete lack of aerosol particles in the atmospheric circulation is expected, since it guarantees that the only mechanism of atmospheric ions loss is the ion-ion recombination. As the concentration of aerosols increases, additional loss can be due to ions attaching to the particles, which leads to a perturbation of the ion density from fair weather values. In actual conditions, aerosols always exist, but under fair weather conditions their concentrations are small enough to not significantly affect the ionic content of the atmosphere. Therefore, for the modelling purposes of fair weather conditions, aerosol concentrations can be neglected."

**References:**

Siingh, D., Gopalakrishnan, V., Singh, R. P., Kamra, A. K., Singh, S., Pant, V., Singh, R. and Singh, A. K.: The atmospheric global electric circuit: An overview, Atmos. Res., 84(2), 91–110, doi:10.1016/j.atmosres.2006.05.005, 2007.

Kastelis, N. and Kourtidis, K.: Characteristics of the atmospheric electric field and correlation with CO 2 at a rural site in southern Balkans, Earth, Planets Sp., doi:10.1186/s40623-016-0379-3, 2016.

*Line 209" near ground atmospheric conductivity and the atmospheric scale height...": So, you assume _ to be proportional to density, I guess. Perhaps you could add a line or two on the foundations of this assumption.*

**Author's Response:** We thank the reviewer for this remark and have added the following sentence:

Modified **Line 209 to Lines 233-235**: "The given mathematical formalism of the atmospheric conductivity is adopted also by Ilin et al, 2020. The authors demonstrated that such a profile adequately describes the main aspects of the real conductivity distribution, and can be seen as a global mean conductivity profile."

**Reference** (*not previously included in the manuscript*):

Ilin N.V., Slyunyaev N.N., and Mareev E.A., Towards a realistic representation of global electric circuit generators in models of atmospheric dynamics, J. Geophys. Res. Atmospheres, 125, doi:10.1029/2019JD032130, 2020.

*Line 246: A reference to Instructor's Solution Manual for Introduction to Electrodynamics, 4th Edition, 2013, is needed.*

**Line 246**, updated to **Line 275:** corrected the reference from "Griffiths Instructor's Solutions Manual" to "Griffiths Instructor's Solution Manual for Introduction to Electrodynamics, 4th Edition, 2013".

*Figure 1 could (and should, to my taste) be incorporated into Figs 4-7.*

**Author's Response:** We think that Figures 4 to 7 (later revised to Fig. 6 to 8) are already clustered with information and potential incorporation of the HYSPLIT back trajectories will deter the reader from focusing on the E-field details and would rather highlight the dust layer origin and progression. Under the specific scope, the authors recommend that Figure 1 is kept as is.

*Figs 4 and 5 captions "VLDR values between 25% and 30% indicate the presence of mostly mixed dust ", Figs 6 and 7 captions "VLDR values between 35% and 40% indicate a pure dust layer ": Values >30% but <35% indicating what?*

**Author's Response:** Please refer to the **Line 128-129** previous comment response. We have corrected the VLDR values range for the different aerosol types on all figure captions and text highlights.

---

## Author Comment (AC2) · 24 Nov 2020

**RESPONSE TO REFEREE#2 COMMENTS AND PEER-REVIEW REPORT**

Manuscript Title: **"The Electrical Activity of Saharan Dust as perceived from Surface Electric Field Observations in Greece"**

revised per **reviewer#1** comments as:

**"The Electrical Activity of Saharan Dust as perceived from Surface Electric Field Observations"**

Authors (as declared in the submitted manuscript with an addition of Ms. Ioanna Tsikoudi, justified contribution to the declaration of author contributions):

**Vasiliki Daskalopoulou, Sotirios A. Mallios, Zbigniew Ulanowski, George Hloupis, Anna Gialitaki, Ioanna Tsikoudi, Konstantinos Tassis and Vassilis Amiridis**

Dear respected Editor/Reviewers,

The authors highly appreciate the comprehensive feedback throughout the review process and kindly integrate the reviewer comments in the revised preprint manuscript, as follows:

**REFEREE#2 COMMENTS:**

*"The authors study four dust events in Greece, using a combination of ground-based electric field measurements and lidar. âa The events involve dust that originated in the Sahara 48 to 72 hours previously.âa Two of the events enhance the electric field relative to the reference fair weather field, and the other two events diminish the electric field.*

*This paper presents a simple model to describe these electrical effects.âa There are two components of the model.âa First, that the dust will reduce the conductivity in the region it occupies by scavenging ions; this effect occurs even with neutral dust particles. âa Second, there could be regions of charged dust – this is modeled as cylinders of monopolar charge (there could be two cylinders, one of positive and one of negative charge). Some of the parameters for the model can be obtained from the lidar, while other parameters cannot be independently obtained.*

*Here is where I get lost.Ă ˘a I found the results section very hard to follow.âa It appears to me the authors show experimental results for dust event (Figs 4-7), and then present results of the model under various parameters (Figs 8-13)."*

**Author's Response:** The authors are grateful to the respected reviewer for the constructive comments, up to this point concerning the readability of the manuscript. We agree that the results section appeared perplexed to the first read, therefore we have strongly revised the structure of this section by keeping the synergistic observations between the E-field and the dust layer optical properties, which are the paper's highlighted significance to our perspective, and have given to the physical reasoning behind our findings, through the 1D model outputs, a separate section (section 3). We believe that according to the referee's guidelines this further fortifies the paper and clarifies the distinction the authors attempt to do for lofted electrified dust.
All the structural changes are listed under the respective comment sections in detail.

**Line 258** updated to **Line 288:** Section 3 from "Results" becomes "Model outputs", where we keep only the model results for the different cylinder configurations.

**Section 3** is now subdivided to the following paragraphs:
  - **Lines 289 – 295:** introductory paragraph, "As a result of the mathematical formalism…distances."

- **Lines 323 – 333 & 346 – 352** become **Lines 296 – 311:** section **3.1 E-field below Fair weather field**, "In this section, we describe the possible cases under which lofted dust layers...lidar PLDR profiles (Table 1)."
- **Lines 353 – 390** modified to **Lines 312 – 346:** section **3.1.1 Balanced/Imbalanced dipole field below Fair weather**, "We consider the case of two oppositely charged cylinders …below the fair weather value"
- **Lines 391 – 393** become **Lines 347 – 349:** section **3.2 E-field above Fair weather field**, "We examine…the LREF."
- **Lines 394 – 412** modified to **Lines 350 – 360:** section **3.2.1 Balanced/Imbalanced dipole field above Fair weather field**, "For the same…location (below point A)."

**Section 4** named "Experimental Results" and is subdivided to:
- **Lines 259 – 261** modified to **Lines 343 – 366:** introductory paragraphs to the section, "The near ground electric… supported by the model configuration described in the previous sections."
- **Lines 262 – 270** modified to **Lines 370 – 388:** section **4.1 Dust Layer characterization through lidar**, "The July 2017 and… very low concentrations of dust particles are also present within the MABL."
- **Lines 271 – 279** modified to **Lines 389 - 399:** section **4.2 Local mean E-field behaviour**, "Considering the electrical properties… the expected fair weather value."
- **Lines 283 – 295** become **Lines 400 – 414: s**ection **4.3 Observed E-field enhancement as compared to LREF**, "In Fig. 6 and Fig. 7… for even smaller charge separation distances."
- **Lines 296 – 312** become **Lines 415 – 431:** section **4.4 Observed E-field reduction as compared to LREF**, "Several dust load cases…of the dust layer aloft."
- **Lines 444 – 450** become **Lines 432 - 438:** section **4.5 Reversed E-field polarity**, "If a reversed polarity…to be electrified."

**Section 5** becomes the "Discussion" section and is subdivided to:
- **Lines 413 – 443** section 3.4.3 becomes **Lines 440 – 467** section **5.1 "E-field dependence on the bottom charged are height"**, kept as was in previous manuscript only section numbering changes.
- **Lines 452 – 463** become **Lines 471 – 482** section **5.2 Chauvenet criterion validity**, and is kept the same as in the previous manuscript, only section numbering changes.
- **Lines 464 – 477** become **Lines 483 – 496 s**ection **5.3 Generalization of the cylindrical model and the LREF methodology** and is kept the same as in the previous manuscript, only section numbering changes.

**Lines 497 – 518:** **Section 6** as "Summary and Conclusions", no changes only different numbering

*"I do not think there is much interest in the results of the simple model under various parameters.áa I think these figures and the associated text should be removed. Rather, I think they should focus (succinctly) on using the model to rationalize the experimental results.áa This must be done much better in order for the paper to be publishable."*

**Author's Response:** In this study, we also focus on the solution of the problem of charged cylindrical areas placed within a conducting medium, in order to simulate the charge structure within elevated dust layers and deduce the E-field behavior near the sensor. Since there is a lack of vertical mapping of these structures and only a recent study by Zhang and Zhou, 2020, attempted to construct a map of the electrical structure of dust storms through surface observations, we have selected this formalism to better represent our experimental findings and used it as a proof of concept for our conclusions. The model is indeed a simplistic approach to the structure of charged areas and is based on previously used geometries on cloud electrification, but never before on dust layers. Its purpose is to give physical insights regarding the parameters that influence the electrical properties of dust layers, rather than reproducing the measurements.

Therefore, we believe that it is imperative to maintain the model outputs that are used to rationalize the E-field behavior (previously Figures 9 to 12), but to comply with the referee's kind comments and to diminish the size of relevant information within the main text, we have condensed the information of Fig. 9 – 10 to a single figure, now Fig. 4 (a & b) and that concerning the reduction of the E-field below the fair weather values contained in Fig. 11 – 12, to Fig. 5 (a & b) in section 3/Model outputs. Moreover, the authors consider the sensitivity study of the ground E-field to the basic parameters given in Eq. 11 (previously section 3.3.1 Dust layer acting as a passive element, Fig. 8) to be redundant for the main text, but an important addition as to why the conductivity reduction factor plays a significant

role in the E-field effects and the selection of n based on experimental data. As a result, the respective section is removed from the text and added to Appendix A.

More analytically the changes are:

**Lines 319 – 322:** "Following this formulation…separation distances" updated to **Lines 291 – 295**, section Model outputs.
**Lines 355 – 345:** added to **Appendix A** updated to **Lines 520 – 353**, as "Dust layer acting as a passive element"

**Lines 280 – 282:** "According to the effect…to the local field." updated to **Lines 364 – 3367** and added the phrase "Through these observations we attempt to provide evidence of electrically active dust only by ground-based methods."
**updated Lines 426 – 430:** added the following paragraph "Following the 1D model outputs for such a case (see Section 3.1.1), this observed reduction could be attributed to either electrically neutral dust aloft or to electrically active dust with the charged regions in relatively small separation distances within the layer. Under the electrically active dust case, a charge imbalance of less than 10%, can be adequate to interpret the observed reduction of the E-field below the LREF for even smaller separation distances. But the detection of such an E-field reduction below the LREF cannot conclusively characterize the electrical activity of the dust layer aloft."

**updated Lines 458 – 459:** added the phrase "As such, observations of enhanced E-field above the fair weather values, for dust driven days, can be reproduced only when an electrically active dust layer is transported above the fieldmill."

*"And overall, I think the paper needs to be communicated much more clearly, and walk the reader through the results and the logic behind their ideas.¢a Figure captions should clarify what the data represents (cannot assume someone knows this). ˘a As I said above, I got lost and couldn't understand things."*

**Author's Response:** We would like to think that the present format of the paper is much more reader friendly and easy to follow from the wide scientific community of ACP.

[revised manuscript text omitted]